# Noise-Tolerant Life-Long Matrix Completion via Adaptive Sampling

**Maria-Florina Balcan**
Machine Learning Department
Carnegie Mellon University, USA
ninamf@cs.cmu.edu

**Hongyang Zhang**
Machine Learning Department
Carnegie Mellon University, USA
hongyanz@cs.cmu.edu

## Abstract

We study the problem of recovering an incomplete $m \times n$ matrix of rank $r$ with columns arriving online over time. This is known as the problem of *life-long matrix completion*, and is widely applied to recommendation system, computer vision, system identification, etc. The challenge is to design provable algorithms tolerant to a large amount of noises, with small sample complexity. In this work, we give algorithms achieving strong guarantee under two realistic noise models. In bounded deterministic noise, an adversary can add any bounded yet unstructured noise to each column. For this problem, we present an algorithm that returns a matrix of a small error, with sample complexity almost as small as the best prior results in the noiseless case. For sparse random noise, where the corrupted columns are sparse and drawn randomly, we give an algorithm that *exactly* recovers an $\mu_0$-incoherent matrix by probability at least $1 - \delta$ with sample complexity as small as $\mathcal{O}\left(\mu_0 rn \log(r/\delta)\right)$. This result advances the state-of-the-art work and matches the lower bound in a worst case. We also study the scenario where the hidden matrix lies on a mixture of subspaces and show that the sample complexity can be even smaller. Our proposed algorithms perform well experimentally in both synthetic and real-world datasets.

## 1 Introduction

Life-long learning is an emerging object of study in machine learning, statistics, and many other domains [2, 11]. In machine learning, study of such a framework has led to significant advances in learning systems that continually learn many tasks over time and improve their ability to learn as they do so, like humans [15]. A natural approach to achieve this goal is to exploit information from previously-learned tasks under the belief that some commonalities exist across the tasks [2, 24]. The focus of this work is to apply this idea of life-long learning to the matrix completion problem. That is, given columns of a matrix that arrive online over time with missing entries, how to approximately/exactly recover the underlying matrix by exploiting the low-rank commonality across each column.

Our study is motivated by several promising applications where life-long matrix completion is applicable. In recommendation systems, the column of the hidden matrix consists of ratings by multiple users to a specific movie/news; The news or movies are updated online over time but usually only a few ratings are submitted by those users. In computer vision, inferring camera motion from a sequence of online arriving images with missing pixels has received significant attention in recent years, known as the structure-from-motion problem; Recovering those missing pixels from those partial measurements is an important preprocessing step. Other examples where our technique is applicable include system identification, multi-class learning, global positioning of sensors, etc.

Despite a large amount of applications of life-long matrix completion, many fundamental questions remain unresolved. One of the long-standing challenges is designing *noise-tolerant, life-long*

algorithms that can recover the unknown target matrix with small error. In the absence of noise, this problem is not easy because the overall structure of the low rankness is unavailable in each round. This problem is even more challenging in the context of noise, where an adversary can add any bounded yet unstructured noise to those observations and the error propagates as the algorithm proceeds. This is known as bounded deterministic noise. Another type of noise model that receives great attention is sparse random noise, where the noise is sparse compared to the number of columns and is drawn i.i.d. from a non-degenerate distribution.

**Our Contributions:** This paper tackles the problem of *noise-tolerant, life-long* matrix completion and advances the state-of-the-art results under the two realistic noise models.

- Under bounded deterministic noise, we design and analyze an algorithm that is robust to noise, with only a small output error (See Figure 3). The sample complexity is almost as small as the best prior results in the noiseless case, provided that the noise level is small.

- Under sparse random noise, we give sample complexity that guarantees an *exact* recovery of the hidden matrix with high probability. The sample complexity advances the state-of-the-art results (See Figure 3) and matches the lower bound in the worst case of this scenario.

- We extend our result of sparse random noise to the setting where the columns of the hidden matrix lie on a mixture of subspaces, and show that smaller sample complexity suffices to exactly recover the hidden matrix in this more benign setting.

- We also show that our proposed algorithms perform well experimentally in both synthetic and real-world datasets.

## 2 Preliminaries

Before proceeding, we define some notations and clarify problem setup in this section.

**Notations:** We will use bold capital letter to represent matrix, bold lower-case letter to represent vector, and lower-case letter to represent scalar. Specifically, we denote by $\mathbf{M} \in \mathbb{R}^{m \times n}$ the noisy observation matrix in hindsight. We denote by $\mathbf{L}$ the underlying clean matrix, and by $\mathbf{E}$ the noise. We will frequently use $\mathbf{M}_{:t} \in \mathbb{R}^{m \times 1}$ to indicate the $t$-th column of matrix $\mathbf{M}$, and similarly $\mathbf{M}_{t:} \in \mathbb{R}^{1 \times n}$ the $t$-th row. For any set of indices $\Omega$, $\mathbf{M}_{\Omega:} \in \mathbb{R}^{|\Omega| \times n}$ represents subsampling the rows of $\mathbf{M}$ at coordinates $\Omega$. Without confusion, denote by $\mathbf{U}$ the column space spanned by the matrix $\mathbf{L}$. Denote by $\widetilde{\mathbf{U}}$ the noisy version of $\mathbf{U}$, i.e., the subspace corrupted by the noise, and by $\widehat{\mathbf{U}}$ our estimated subspace. The superscript $k$ of $\widetilde{\mathbf{U}}^k$ means that $\widetilde{\mathbf{U}}^k$ has $k$ columns in the current round. $\mathcal{P}_{\mathbf{U}}$ is frequently used to represent the orthogonal projection operator onto subspace $\mathbf{U}$. We use $\theta(\mathbf{a}, \mathbf{b})$ to denote the angle between vectors $\mathbf{a}$ and $\mathbf{b}$. For a vector $\mathbf{u}$ and a subspace $\mathbf{V}$, define $\theta(\mathbf{u}, \mathbf{V}) = \min_{\mathbf{v} \in \mathbf{V}} \theta(\mathbf{u}, \mathbf{v})$. We define the angle between two subspaces $\mathbf{U}$ and $\mathbf{V}$ as $\theta(\mathbf{U}, \mathbf{V}) = \max_{\mathbf{u} \in \mathbf{U}} \theta(\mathbf{u}, \mathbf{V})$. For norms, denote by $\|\mathbf{v}\|_2$ the vector $\ell_2$ norm of $\mathbf{v}$. For matrix, $\|\mathbf{M}\|_F^2 = \sum_{ij} \mathbf{M}_{ij}^2$ and $\|\mathbf{M}\|_{\infty,2} = \max_i \|\mathbf{M}_{i:}\|_2$, i.e., the maximum vector $\ell_2$ norm across rows. The operator norm is induced by the matrix Frobenius norm, which is defined as $\|\mathcal{P}\| = \max_{\|\mathbf{M}\|_F \leq 1} \|\mathcal{P}\mathbf{M}\|_F$. If $\mathcal{P}$ can be represented as a matrix, $\|\mathcal{P}\|$ also denotes the maximum singular value.

### 2.1 Problem Setup

In the setting of life-long matrix completion, we assume that each column of the underlying matrix $\mathbf{L}$ is normalized[1] and arrives online over time. We are not allowed to get access to the next column until we perform the completion for the current one. This is in sharp contrast to the offline setting where all columns come at one time and so we are able to immediately exploit the low-rank structure to do the completion. In hindsight, we assume the underlying matrix is of rank $r$. This assumption enables us to represent $\mathbf{L}$ as $\mathbf{L} = \mathbf{US}$, where $\mathbf{U}$ is the dictionary (a.k.a. basis matrix) of size $m \times r$ with each column representing a latent metafeature, and $\mathbf{S}$ is a matrix of size $r \times n$ containing the weights of linear combination for each column $\mathbf{L}_{:t}$. The overall subspace structure is captured by $\mathbf{U}$ and the finer grouping structure, e.g., the mixture of multiple subspaces, is captured by the sparsity of $\mathbf{S}$. Our goal is to approximately/exactly recover the subspace $\mathbf{U}$ and the matrix $\mathbf{L}$ from a small fraction of the entries, possibly corrupted by noise, although these entries can be selected sequentially in a feedback-driven way.

**Noise Models:** We study two types of realistic noise models, one of which is the deterministic noise. In this setting, we assume that the $\ell_2$ norm of noise on each column is bounded by $\epsilon_{noise}$. Beyond

that, no other assumptions are made on the nature of noise. The challenge under this noise model is to design an *online* algorithm limiting the possible error propagation during the completion procedure. Another noise model we study is the sparse random noise, where we assume that the noise vectors are drawn i.i.d. from *any* non-degenerate distribution. Additionally, we assume the noise is sparse, i.e., only a few columns of $\mathbf{L}$ are corrupted by noise. Our goal is to *exactly* recover the underlying matrix $\mathbf{L}$ with sample complexity as small as possible.

**Incoherence:** Apart from the sample budget and noise level, another quantity governing the difficulty of the completion problem is the coherence parameter on the row/column space. Intuitively, the completion should perform better when the information spreads evenly throughout the matrix. To quantify this term, for subspace $\mathbf{U}$ of dimension $r$ in $\mathbb{R}^m$, we define

$$\mu(\mathbf{U}) = \frac{m}{r} \max_{i \in [m]} \|\mathcal{P}_{\mathbf{U}}\mathbf{e}_i\|_2^2, \tag{1}$$

where $\mathbf{e}_i$ is the $i$-th column of the identity matrix. Indeed, without (1) there is an identifiability issue in the matrix completion problem [7, 8, 27]. As an extreme example, let $\mathbf{L}$ be a matrix with only one non-zero entry. Such a matrix cannot be exactly recovered unless we see the non-zero element. As in [19], to mitigate the issue, in this paper we assume incoherence $\mu_0 = \mu(\mathbf{U})$ on the column space of the underlying matrix. This is in contrast to the classical results of Candès et al. [7, 8], in which one requires incoherence $\mu_0 = \max\{\mu(\mathbf{U}), \mu(\mathbf{V})\}$ on both the column and the row subspaces.

**Sampling Model:** Instead of sampling the entries passively by uniform distribution, our sampling oracle allows adaptively measuring entries in each round. Specifically, for any arriving column we are allowed to have two types of sampling phases: we can either uniformly take the samples of the entries, as the passive sampling oracle, or choose to request all entries of the column in an adaptive manner. This is a natural extension of the classical passive sampling scheme with wide applications. For example, in network tomography, a network operator is interested in inferring latencies between hosts while injecting few packets into the network. The operator is in control of the network, thus can adaptively sample the matrix of pair-wise latencies. In particular, the operator can request full columns of the matrix by measuring one host to all others. In gene expression analysis, we are interested in recovering a matrix of expression levels for various genes across a number of conditions. The high-throughput microarrays provide expression levels of all genes of interest across operating conditions, corresponding to revealing entire columns of the matrix.

## 3 Main Results

In this section, we formalize our life-long matrix completion algorithm, develop our main theoretical contributions, and compare our results with the prior work.

### 3.1 Bounded Deterministic Noise

To proceed, our algorithm streams the columns of noisy $\mathbf{M}$ into memory and iteratively updates the estimate for the column space of $\mathbf{L}$. In particular, the algorithm maintains an estimate $\widehat{\mathbf{U}}$ of subspace $\mathbf{U}$, and when processing an arriving column $\mathbf{M}_{:t}$, requests only a few entries of $\mathbf{M}_{:t}$ and a few rows of $\widehat{\mathbf{U}}$ to estimate the distance between $\mathbf{L}_{:t}$ and $\mathbf{U}$. If the value of the estimator is greater than a given threshold $\eta_k$, the algorithm requests the remaining entries of $\mathbf{M}_{:t}$ and adds the new direction $\mathbf{M}_{:t}$ to the subspace estimate; Otherwise, finds a best approximation of $\mathbf{M}_{:t}$ by a linear combination of columns of $\widehat{\mathbf{U}}$. The pseudocode of the procedure is displayed in Algorithm 1. We note that our algorithm is similar to the algorithm of [19] for the problem of offline matrix completion without noise. However, our setting, with the presence of noise (which might conceivably propagate through the course of the algorithm), makes our analysis significantly more subtle.

The key ingredient of the algorithm is to estimate the distance between the noiseless column $\mathbf{L}_{:t}$ and the clean subspace $\mathbf{U}^k$ with only a few measurements with noise. To estimate this quantity, we downsample both $\mathbf{M}_{:t}$ and $\widehat{\mathbf{U}}^k$ to $\mathbf{M}_{\Omega t}$ and $\widehat{\mathbf{U}}^k_{\Omega:}$, respectively. We then project $\mathbf{M}_{\Omega t}$ onto subspace $\widehat{\mathbf{U}}^k_{\Omega:}$ and use the projection residual $\|\mathbf{M}_{\Omega t} - \mathcal{P}_{\widehat{\mathbf{U}}^k_{\Omega:}}\mathbf{M}_{\Omega t}\|_2$ as our estimator. A subtle and critical aspect of the algorithm is the choice of the threshold $\eta_k$ for this estimator. In the noiseless setting, we can simply set $\eta_k = 0$ if the sampling number $|\Omega|$ is large enough — in the order of $\mathcal{O}(\mu_0 r \log^2 r)$, because $\mathcal{O}(\mu_0 r \log^2 r)$ noiseless measurements already contain enough information for testing whether a specific column lies in a given subspace [19]. In the noisy setting, however, the

---
**Algorithm 1** Noise-Tolerant Life-Long Matrix Completion under Bounded Deterministic Noise
---
    **Input:** Columns of matrices arriving over time.

    **Initialize:** Let the basis matrix $\widehat{\mathbf{U}}^0 = \emptyset$. Randomly draw entries $\Omega \subset [m]$ of size $d$ uniformly with replacement.

    **1: For** $t$ from 1 to $n$, **do**

    **2:**     (a) If $\|\mathbf{M}_{\Omega t} - \mathcal{P}_{\widehat{\mathbf{U}}^k_{\Omega:}} \mathbf{M}_{\Omega t}\|_2 > \eta_k$

    **3:**        i. Fully measure $\mathbf{M}_{:t}$ and add it to the basis matrix $\widehat{\mathbf{U}}^k$. Orthogonalize $\widehat{\mathbf{U}}^k$.

    **4:**        ii. Randomly draw entries $\Omega \subset [m]$ of size $d$ uniformly with replacement.

    **5:**        iii. $k := k + 1$.

    **6:**     (b) Otherwise $\widehat{\mathbf{M}}_{:t} := \widehat{\mathbf{U}}^k \widehat{\mathbf{U}}^{k\dagger}_{\Omega:} \mathbf{M}_{\Omega t}$.

    **7: End For**

    **Output:** Estimated range space $\widehat{\mathbf{U}}^K$ and the underlying matrix $\widehat{\mathbf{M}}$ with column $\widehat{\mathbf{M}}_{:t}$.
---

challenge is that both $\mathbf{M}_{:t}$ and $\widehat{\mathbf{U}}^k$ are corrupted by noise, and the error propagates as the algorithm proceeds. Thus instead of setting the threshold as 0 always, our theory suggests setting $\eta_k$ proportional to the noise level $\sqrt{\epsilon_{noise}}$. Indeed, the threshold $\eta_k$ balances the trade-off between the estimation error and the sample complexity: a) if $\eta_k$ is too large, most of the columns are represented by the noisy dictionary and therefore the error propagates too quickly; b) In contrast, if $\eta_k$ is too small, we observe too many columns in full and so the sample complexity increases. Our goal in this paper is to capture this trade-off, providing a global upper bound on the estimation error of the life-long arriving columns while keeping the sample complexity as small as possible.

### 3.1.1 Recovery Guarantee

Our analysis leads to the following guarantee on the performance of Algorithm 1.

**Theorem 1** (Robust Recovery under Deterministic Noise). *Let $r$ be the rank of the underlying matrix $\mathbf{L}$ with $\mu_0$-incoherent column space. Suppose that the $\ell_2$ norm of noise in each column is upper bounded by $\epsilon_{noise}$. Set the parameters $d \geq c(\mu_0 r + mk\epsilon_{noise}) \log^2(2n/\delta))$ and $\eta_k = C\sqrt{dk\epsilon_{noise}/m}$ for global constants $c$ and $C$. Then with probability at least $1 - \delta$, Algorithm 1 outputs $\widehat{\mathbf{U}}^K$ with $K \leq r$ and outputs $\widehat{\mathbf{M}}$ with $\ell_2$ error $\|\widehat{\mathbf{M}}_{:t} - \mathbf{L}_{:t}\|_2 \leq \mathcal{O}\left(\frac{m}{d}\sqrt{k\epsilon_{noise}}\right)^2$ uniformly for all $t$, where $k \leq r$ is the number of base vectors when processing the $t$-th column.*

*Proof Sketch.* We firstly show that our estimated subspace in each round is accurate. The key ingredient of our proof is a result pertaining the angle between the underlying subspace and the noisy one. Ideally, the column space spanned by the noisy dictionary cannot be too far to the underlying subspace if the noise level is small. This is true only if *the angle between the newly added vector and the column space of the current dictionary is large*, as shown by the following lemma.

**Lemma 2.** *Let $\mathbf{U}^k = \mathbf{span}\{\mathbf{u}_1, \mathbf{u}_2, ..., \mathbf{u}_k\}$ and $\widetilde{\mathbf{U}}^k = \mathbf{span}\{\widetilde{\mathbf{u}}_1, \widetilde{\mathbf{u}}_2, ..., \widetilde{\mathbf{u}}_k\}$ be two subspaces such that $\theta(\mathbf{u}_i, \widetilde{\mathbf{u}}_i) \leq \epsilon_{noise}$ for all $i \in [k]$. Let $\gamma_k = \sqrt{20k\epsilon_{noise}}$ and $\theta(\widetilde{\mathbf{u}}_i, \widetilde{\mathbf{U}}^{i-1}) \geq \gamma_i$ for $i = 2, ..., k$. Then $\theta(\mathbf{U}^k, \widetilde{\mathbf{U}}^k) \leq \gamma_k/2$.*

We then prove the correctness of our test in Step 2. Lemma 2 guarantees that the underlying subspace $\mathbf{U}^k$ and our estimated one $\widetilde{\mathbf{U}}^k$ cannot be too distinct. So by algorithm, projecting any vector on the subspace spanned by $\widetilde{\mathbf{U}}^k$ does not make too many mistakes, i.e., $\theta(\mathbf{M}_{:t}, \widetilde{\mathbf{U}}^k) \approx \theta(\mathbf{M}_{:t}, \mathbf{U}^k)$. On the other hand, by standard concentration argument our test statistic $\|\mathbf{M}_{\Omega t} - \mathcal{P}_{\widetilde{\mathbf{U}}^k_{\Omega:}} \mathbf{M}_{\Omega t}\|_2$ is close to $\frac{d}{m}\|\mathbf{M}_{:t} - \mathcal{P}_{\widetilde{\mathbf{U}}^k}\mathbf{M}_{:t}\|_2$. Note that the latter term is determined by the angle of $\theta(\mathbf{M}_{:t}, \widetilde{\mathbf{U}}^k)$. Therefore, our test statistic in Step 2 is indeed an effective measure of $\theta(\mathbf{M}_{:t}, \widetilde{\mathbf{U}}^k)$, or $\theta(\mathbf{L}_{:t}, \widetilde{\mathbf{U}}^k)$ since $\mathbf{L}_{:t} \approx \mathbf{M}_{:t}$, as proven by the following novel result.

**Lemma 3.** *Let $\epsilon_k = 2\gamma_k$, $\gamma_k = \sqrt{20k\epsilon_{noise}}$, and $k \leq r$. Suppose that we observe a set of coordinates $\Omega \subset [m]$ of size $d$ uniformly at random with replacement, where $d \geq c_0(\mu_0 r + mk\epsilon_{noise}) \log^2(2/\delta)$. If $\theta(\mathbf{L}_{:t}, \widetilde{\mathbf{U}}^k) \leq \epsilon_k$, then with probability at least $1 - 4\delta$, we have $\|\mathbf{M}_{\Omega t} - \mathcal{P}_{\widetilde{\mathbf{U}}^k_{\Omega:}} \mathbf{M}_{\Omega t}\|_2 \leq C\sqrt{dk\epsilon_{noise}/m}$. Inversely, if $\theta(\mathbf{L}_{:t}, \widetilde{\mathbf{U}}^k) \geq c\epsilon_k$, then with probability at least $1 - 4\delta$, we have $\|\mathbf{M}_{\Omega t} - \mathcal{P}_{\widetilde{\mathbf{U}}^k_{\Omega:}} \mathbf{M}_{\Omega t}\|_2 \geq C\sqrt{dk\epsilon_{noise}/m}$, where $c_0$, $c$ and $C$ are absolute constants.*

Finally, as both our dictionary and our statistic are accurate, the output error cannot be too large. A simple deduction on the union bound over all columns leads to Theorem 1. □

Theorem 1 implies a result in the noiseless setting when $\epsilon_{noise}$ goes to zero. Indeed, with the sample size growing in the order of $\mathcal{O}(\mu_0 nr \log^2 n)$, Algorithm 1 outputs a solution that is exact with probability at least $1 - \frac{1}{n^{10}}$. To the best of our knowledge, this is the best sample complexity in the existing literature for noiseless matrix completion without additional side information [19, 22]. For the noisy setting, Algorithm 1 enjoys the same sample complexity $\mathcal{O}(\mu_0 nr \log^2 n)$ as the noiseless case, if $\epsilon_{noise} \leq \Theta(\mu_0 r/(mk))$. In addition, Algorithm 1 inherits the benefits of adaptive sampling scheme. The vast majority results in the passive sampling scenarios require both the row and column incoherence for exact/robust recovery [22]. In contrast, via adaptive sampling we can relax the incoherence assumption on the row space of the underlying matrix and are therefore more applicable.

We compare our result with several related lines of research in the prior work. While lots of online matrix completion algorithms have been proposed recently, they either lack of solid theoretical guarantee [17], or require strong assumptions for the streaming data [19, 21, 13, 18]. Specifically, Krishnamurthy et al. [18] proposed an algorithm that requires column subset selection in the noisy case, which might be impractical in the online setting as we cannot measure columns that do not arrive. Focusing on a similar online matrix completion problem, Lois et al. [21] assumed that a) there is a good initial estimate for the column space; b) the column space changes slowly; c) the base vectors of the column space are dense; d) the support of the measurements changes by at least a certain amount. In contrast, our assumptions are much simpler and more realistic.

We mention another related line of research — matched subspace detection. The goal of matched subspace detection is to decide whether an incomplete signal/vector lies within a given subspace [5, 4]. It is highly related to the procedure of our algorithm in each round, where we aim at determining whether an arriving vector belongs to a given subspace based on partial and noisy observations. Prior work targeting on this problem formalizes the task as a hypothesis testing problem. So they assume a specific random distribution on the noise, e.g., Gaussian, and choose $\eta_k$ by fixing the probability of false alarm in the hypothesis testing [5, 23]. Compared with this, our result does not have any assumption on the noise structure/distribution.

## 3.2 Sparse Random Noise

In this section, we discuss life-long matrix completion on a simpler noise model but with a stronger recovery guarantee. We assume that noise is sparse, meaning that the total number of noisy columns is small compared to the total number of columns $n$. The noisy columns may arrive at any time, and each noisy column is assumed to be drawn i.i.d. from a non-degenerate distribution. Our goal is to *exactly* recover the underlying matrix and identify the noise with high probability.

We use an algorithm similar to Algorithm 1 to attack the problem, with $\eta_k = 0$. The challenge is that here we frequently add noise vectors to the dictionary and so we need to distinguish the noise from the clean column and remove them out of the dictionary at the end of the algorithm. To resolve the issue, we additionally record the support of the representation coefficients in each round when we represent the arriving vector by the linear combinations of the columns in the dictionary matrix. On one hand, the noise vectors in the dictionary fail to represent any column, because they are random. So if the representation coefficient corresponding to a column in the dictionary is 0 always, it is convincing to identify the column as a noise. On the other hand, to avoid recognizing a true base vector as a noise, we make a mild assumption that the underlying column space is identifiable. Typically, that means for each direction in the underlying subspace, there are at least two clean data points having non-zero projection on that direction. We argue that the assumption is indispensable, since without it there is an identifiability issue between the clean data and the noise. As an extreme example, we cannot identify the black point in Figures 1 as the clean data or as noise if we make no assumption on the underlying subspace. To mitigate the problem, we assume that for each $i \in [r]$ and a subspace $\mathbf{U}^r$ with orthonormal basis, there are at least two columns $\mathbf{L}_{:a_i}$ and $\mathbf{L}_{:b_i}$ of $\mathbf{L}$ such that $[\mathbf{U}^r]_{:i}^T \mathbf{L}_{:a_i} \neq 0$ and $[\mathbf{U}^r]_{:i}^T \mathbf{L}_{:b_i} \neq 0$. The detailed algorithm can be found in the supplementary material.

### 3.2.1 Upper Bound

We now provide upper and lower bound on the sample complexity of above algorithm for the exact recovery of underlying matrix. Our upper bound matches the lower bound up to a constant factor. We then analyze a more benign setting, namely, the data lie on a mixture of low-rank subspaces with

Table 1: Comparisons of our sample complexity with the best prior results in the noise-free setting.

| | Passive Sampling | Adaptive Sampling | |
|---|---|---|---|
| Complexity | $\mathcal{O}\left(\mu_0 nr \log^2(n/\delta)\right)$[22] | $\mathcal{O}\left(\mu_0 nr \log^2(r/\delta)\right)$[19] | $\mathcal{O}\left(\mu_0 nr \log(r/\delta)\right)$ (Ours) |
| Lower bound | $\mathcal{O}\left(\mu_0 nr \log(n/\delta)\right)$[10] | $\mathcal{O}\left(\mu_0 nr \log(r/\delta)\right)$ (Ours) | |

dimensionality $\tau \ll r$. Our analysis leads to the following guarantee on the performance of above algorithm. The proof is in the supplementary material.

**Theorem 4** (Exact Recovery under Random Noise). *Let $r$ be the rank of the underlying matrix $\mathbf{L}$ with $\mu_0$-incoherent column space. Suppose that the noise $\mathbf{E}^{s_0}$ of size $m \times s_0$ are drawn from any non-degenerate distribution, and that the underlying subspace $\mathbf{U}^r$ is identifiable. Then our algorithm exactly recovers the underlying matrix $\mathbf{L}$, the column space $\mathbf{U}^r$, and the outlier $\mathbf{E}^{s_0}$ with probability at least $1 - \delta$, provided that $d \geq c\mu_0 r \log(r/\delta)$ and $s_0 \leq d - r - 1$. The total sample complexity is thus $c\mu_0 rn \log(r/\delta)$, where $c$ is a universal constant.*

Theorem 4 implies an immediate result in the noise-free setting as $\epsilon_{noise}$ goes to zero. In particular, $\mathcal{O}(\mu_0 nr \log(r/\delta))$ measurements are sufficient so that our algorithm outputs a solution that is exact with probability at least $1 - \delta$. This sample complexity improves over existing results of $\mathcal{O}\left(\mu_0 nr \log^2(n/\delta)\right)$ [22] and $\mathcal{O}\left(\mu_0 nr^{3/2} \log(r/\delta)\right)$ [18], and over $\mathcal{O}\left(\mu_0 nr \log^2(r/\delta)\right)$ of Theorem 1 when $\epsilon_{noise} = 0$. Indeed, our sample complexity $\mathcal{O}(\mu_0 nr \log(r/\delta))$ matches the lower bound, as shown by Theorem 5 (See Table 1 for comparisons of sample complexity). We notice another paper of Gittens [14] which showed that Nsytröm method recovers a *positive-semidefinite* matrix of rank $r$ from uniformly sampling $\mathcal{O}(\mu_0 r \log(r/\delta))$ columns. While this result matches our sample complexity, the assumptions of positive-semidefiniteness and of subsampling the columns are impractical in the online setting.

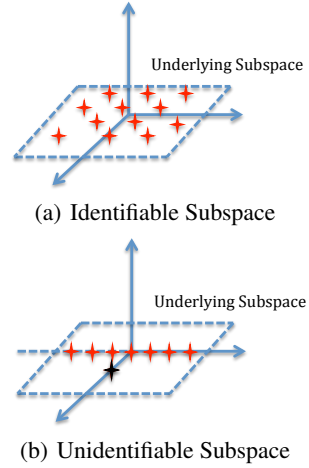

Underlying Subspace

(a) Identifiable Subspace

Underlying Subspace

(b) Unidentifiable Subspace

Figure 1: Identifiability.

We compare Theorem 4 with prior methods on decomposing an incomplete matrix as the sum of a low-rank term and a column-sparse term. Probably one of the best known algorithms is Robust PCA via Outlier Pursuit [25, 28, 27, 26]. Outlier Pursuit converts this problem to a convex program:

$$\min_{\mathbf{L},\mathbf{E}} \|\mathbf{L}\|_* + \lambda \|\mathbf{E}\|_{2,1}, \;\; \text{s.t.} \;\; \mathcal{P}_\Omega \mathbf{M} = \mathcal{P}_\Omega(\mathbf{L} + \mathbf{E}), \tag{2}$$

where $\|\cdot\|_*$ captures the low-rankness of the underlying subspace and $\|\cdot\|_{2,1}$ captures the column-sparsity of the noise. Recent papers on Outlier Pursuit [26] prove that the solution to (2) exactly recovers the underlying subspace, provided that $d \geq c_1 \mu_0^2 r^2 \log^3 n$ and $s_0 \leq c_2 d^4 n/(\mu_0^5 r^5 m^3 \log^6 n)$ for constants $c_1$ and $c_2$. Our result definitely outperforms the existing result in term of the sample complexity $d$, while our dependence of $s_0$ is not always better (although in some cases better) when $n$ is large. Note that while Outlier Pursuit loads all columns simultaneously and so can exploit the global low-rank structure, our algorithm is online and therefore cannot tolerate too much noise.

### 3.2.2 Lower Bound

We now establish a lower bound on the sample complexity. Our lower bound shows that in our adaptive sampling setting, one needs at least $\Omega(\mu_0 rn \log(r/\delta))$ many samples in order to uniquely identify a certain matrix *in the worst case*. This lower bound matches our analysis of upper bound in Section 3.2.1.

**Theorem 5** (Lower Bound on Sample Complexity). *Let $0 < \delta < 1/2$, and $\Omega \sim \text{Uniform}(d)$ be the index of the row sampling $\subseteq [m]$. Suppose that $\mathbf{U}^r$ is $\mu_0$-incoherent. If the total sampling number $dn < c\mu_0 rn \log(r/\delta)$ for a constant $c$, then with probability at least $1 - \delta$, there is an example of $\mathbf{M}$ such that under the sampling model of Section 2.1 (i.e., when a column arrives the choices are either (a) randomly sample or (b) view the entire column), there exist infinitely many matrices $\mathbf{L}'$ of rank $r$ obeying $\mu_0$-incoherent condition on column space such that $\mathbf{L}'_{\Omega:} = \mathbf{L}_{\Omega:}$.*

The proof can be found in the supplementary material. We mention several lower bounds on the sample complexity for passive matrix completion. The first is the paper of Candès and Tao [10], that

gives a lower bound of $\Omega(\mu_0 nr \log(n/\delta))$ if the matrix has both incoherent rows and columns. Taking a weaker assumption, Krishnamurthy and Singh [18, 19] showed that if the row space is *coherent*, any passive sampling scheme followed by any recovery algorithm must have $\Omega(mn)$ measurements. In contrast, Theorem 5 demonstrates that in the absence of row-space incoherence, exact recovery of the matrix is possible with only $\Omega(\mu_0 nr \log(r/\delta))$ samples, if the sampling scheme is adaptive.

### 3.2.3   Extension to Mixture of Subspaces

Theorem 5 gives a lower bound on sample complexity in the *worst* case. In this section, we explore the possibility of further reducing the sample complexity with more complex common structure. We assume that the underlying subspace is a mixture of $h$ independent subspaces[3] [20], each of which is of dimension at most $\tau \ll r$. Such an assumption naturally models settings in which there are really $h$ different categories of movies/news while they share a certain commonality across categories. We can view this setting as a network with two layers: The first layer captures the overall subspace with $r$ metafeatures; The second layer is an output layer, consisting of metafeatures each of which is a linear combination of only $\tau$ metafeatures in the first layer. See Figures 2 for visualization. Our argument shows that the sparse connections between the two layers significantly improve the sample complexity.

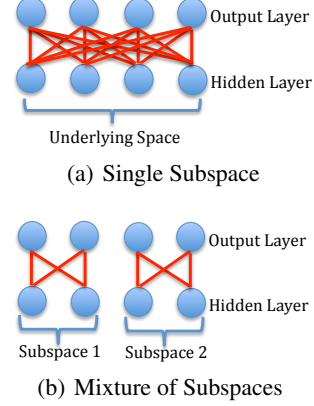

(a) Single Subspace

(b) Mixture of Subspaces

Figure 2: Subspace structure.

Algorithmically, given a new column, we uniformly sample $\tilde{\mathcal{O}}(\tau \log r)$ entries as our observations. We try to represent those elements by a *sparse* linear combination of only $\tau$ columns in the basis matrix, whose rows are truncated to those sampled indices; If we fail, we measure the column in full, add that column into the dictionary, and repeat the procedure for the next arriving column. See supplementary material for the detailed algorithm.

Regarding computational considerations, learning a $\tau$-sparse representation of a given vector w.r.t. a known dictionary can be done in polynomial time if the dictionary matrix satisfies the restricted isometry property [9], or trivially if $\tau$ is a constant [2]. This can be done by applying $\ell_1$ minimization or brute-force algorithm, respectively. Indeed, many real datasets match the constant-$\tau$ assumption, e.g., face image [6] (each person lies on a subspace of dimension $\tau = 9$), 3D motion trajectory [12] (each object lies on a subspace of dimension $\tau = 4$), handwritten digits [16] (each script lies on a subspace of dimension $\tau = 12$), etc. So our algorithm is applicable for all these settings.

Theoretically, the following theorem provides a strong guarantee for our algorithm. The proof can be found in the supplementary material.

**Theorem 6** (Mixture of Subspaces). *Let $r$ be the rank of the underlying matrix $\mathbf{L}$. Suppose that the columns of $\mathbf{L}$ lie on a mixture of identifiable and independent subspaces, each of which is of dimension at most $\tau$. Denote by $\mu_\tau$ the maximal incoherence over all $\tau$-combinations of $\mathbf{L}$. Let the noise model be that of Theorem 4. Then our algorithm exactly recovers the underlying matrix $\mathbf{L}$, the column space $\mathbf{U}^r$, and the outlier $\mathbf{E}^{s_0}$ with probability at least $1 - \delta$, provided that $d \geq c\mu_\tau \tau^2 \log(r/\delta)$ for some global constant $c$ and $s_0 \leq d - \tau - 1$. The total sample complexity is thus $c\mu_\tau \tau^2 n \log(r/\delta)$.*

As a concrete example, if the incoherence parameter $\mu_\tau$ is a global constant and the dimension $\tau$ of each subspace is far less than $r$, the sample complexity of $\mathcal{O}(\mu_\tau n \tau^2 \log(r/\delta))$ is significantly better than the complexity of $\mathcal{O}(\mu_0 nr \log(r/\delta))$ for the structure of a single subspace in Theorem 4. This argument shows that the sparse connections between the two layers improve the sample complexity.

## 4   Experimental Results

**Bounded Deterministic Noise:** We verify the estimated error of our algorithm in Theorem 1 under bounded deterministic noise. Our synthetic data are generated as follows. We construct 5 base vectors $\{\mathbf{u}_i\}_{i=1}^5$ by sampling their entries from $\mathcal{N}(0,1)$. The underlying matrix $\mathbf{L}$ is then generated by $\mathbf{L} = \left[ \mathbf{u}_1 \mathbf{1}_{200}^T, \sum_{i=1}^2 \mathbf{u}_i \mathbf{1}_{200}^T, \sum_{i=1}^3 \mathbf{u}_i \mathbf{1}_{200}^T, \sum_{i=1}^4 \mathbf{u}_i \mathbf{1}_{200}^T, \sum_{i=1}^5 \mathbf{u}_i \mathbf{1}_{1,200}^T \right] \in \mathbb{R}^{100 \times 2,000}$, each column of which is normalized to the unit $\ell_2$ norm. Finally, we add bounded yet unstructured noise to each column, with noise level $\epsilon_{noise} = 0.6$. We randomly pick 20% entries to be unobserved. The left figure in Figure 3 shows the comparison between our estimated error[4] and the true error by our

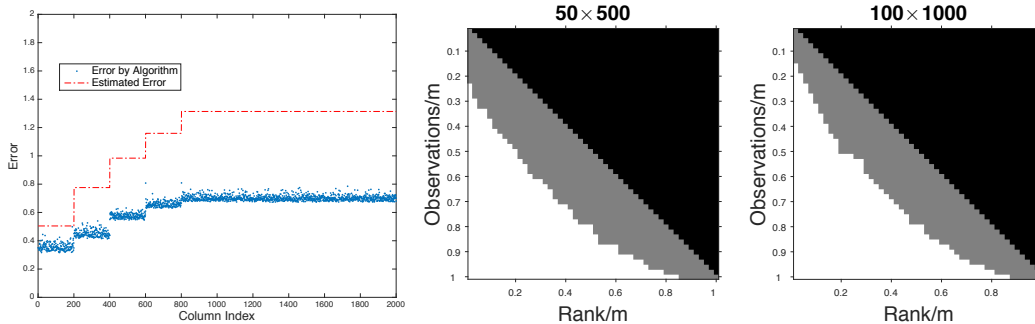

Figure 3: Left Figure: Approximate recovery under bounded deterministic noise with estimated error. Right Two Figures: Exact recovery under sparse random noise with varying rank and sample size. **White Region:** Nuclear norm minimization (passive sampling) succeeds. **White and Gray Regions:** Our algorithm (adaptive sampling) succeeds. **Black Region:** Our algorithm fails. It shows that the success region of our algorithm strictly contains that of the passive sampling method.

algorithm. The result demonstrates that empirically, our estimated error successfully predicts the trend of the true algorithmic error.

**Sparse Random Noise:** We then verify the exact recoverability of our algorithm under sparse random noise. The synthetic data are generated as follows. We construct the underlying matrix $\mathbf{L} = \mathbf{XY}$ as a product of $m \times r$ and $r \times n$ i.i.d. $\mathcal{N}(0,1)$ matrices. The sparse random noise is drawn from standard Gaussian distribution such that $s_0 \leq d - r - 1$. For each size of problem ($50 \times 500$ and $100 \times 1,000$), we test with different rank ratios $r/m$ and measurement ratios $d/m$. The experiment is run by 10 times. We define that the algorithm succeeds if $\|\widehat{\mathbf{L}} - \mathbf{L}\|_F \leq 10^{-6}$, $\mathbf{rank}(\widehat{\mathbf{L}}) = r$, and the recovered support of the noise is exact for at least one experiment. The right two figures in Figure 3 plots the fraction of correct recoveries: white denotes perfect recovery by nuclear norm minimization approach (2); white+gray represents perfect recovery by our algorithm; black indicates failure for both methods. It shows that the success region of our algorithm strictly contains that of the prior approach. Moreover, the phase transition of our algorithm is nearly a linear function w.r.t $r$ and $d$. This is consistent with our prediction $d = \Omega(\mu_0 r \log(r/\delta))$ when $\delta$ is small, e.g., poly$(1/n)$.

**Mixture of Subspaces:** To test the performance of our algorithm for the mixture of subspaces, we conduct an experiment on the Hopkins 155 dataset. The Hopkins 155 database is composed of 155 matrices/tasks, each of which consists of multiple data points drawn from two or three motion objects. The trajectory of each object lie in a subspace. We input the data matrix to our algorithm with varying sample sizes. Table 2 records the average relative error $\|\widehat{\mathbf{L}} - \mathbf{L}\|_F / \|\mathbf{L}\|_F$ of 10 trials for the first five tasks in the dataset. It shows that our algorithm is able to recover the target matrix with high accuracy. Another experiment comparing the sample complexity of single subspace v.s. mixture of subspaces can be found in the supplementary material.

Table 2: Life-long Matrix Completion on the first 5 tasks in Hopkins 155 database.

| #Task | Motion Number | $d = 0.8m$ | $d = 0.85m$ | $d = 0.9m$ | $d = 0.95m$ |
|---|---|---|---|---|---|
| #1 | 2 | $9.4 \times 10^{-3}$ | $6.0 \times 10^{-3}$ | $3.4 \times 10^{-3}$ | $2.6 \times 10^{-3}$ |
| #2 | 3 | $5.9 \times 10^{-3}$ | $4.4 \times 10^{-3}$ | $2.4 \times 10^{-3}$ | $1.9 \times 10^{-3}$ |
| #3 | 2 | $6.3 \times 10^{-3}$ | $4.8 \times 10^{-3}$ | $2.8 \times 10^{-3}$ | $7.2 \times 10^{-4}$ |
| #4 | 2 | $7.1 \times 10^{-3}$ | $6.8 \times 10^{-3}$ | $6.1 \times 10^{-3}$ | $1.5 \times 10^{-3}$ |
| #5 | 2 | $8.7 \times 10^{-3}$ | $5.8 \times 10^{-3}$ | $3.1 \times 10^{-3}$ | $1.2 \times 10^{-3}$ |

## 5   Conclusions

In this paper, we study life-long matrix completion that aims at online recovering an $m \times n$ matrix of rank $r$ under two realistic noise models — bounded deterministic noise and sparse random noise. Our result advances the state-of-the-art work and matches the lower bound under sparse random noise. In a more benign setting where the columns of the underlying matrix lie on a mixture of subspaces, we show that a smaller sample complexity is possible to exactly recover the target matrix. It would be interesting to extend our results to other realistic noise models, including random classification noise or malicious noise previously studied in the context of supervised classification [1, 3]

**Acknowledgements.** This work was supported in part by grants NSF-CCF 1535967, NSF CCF-1422910, NSF CCF-1451177, a Sloan Fellowship, and a Microsoft Research Fellowship.

## Footnotes

[1]Without loss of generality, we assume $\|\mathbf{L}_{:t}\|_2 = 1$ for all $t$, although our result can be easily extended to the general case.

[2]By our proof, the constant factor is 9.

[3] $h$ linear subspaces are independent if the dimensionality of their sum is equal to the sum of their dimensions.

[4] The estimated error is up to a constant factor.

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
