[Supplementary Material]

# Supplementary Material of
# *Noise-Tolerant Life-Long Matrix Completion via Adaptive Sampling*

**Maria-Florina Balcan**
Machine Learning Department
Carnegie Mellon University, USA
ninamf@cs.cmu.edu

**Hongyang Zhang**
Machine Learning Department
Carnegie Mellon University, USA
hongyanz@cs.cmu.edu

## 1 Supplementary Experiments

### 1.1 Single Subspace v.s. Mixture of Subspaces

We compare the sample complexity of Algorithm 2 (Single Subspace) and Algorithm 3 (Mixture of Subspaces) for the exact recovery of the underlying matrix. The data are generated as follows. We construct 5 independent subspaces $\{\mathcal{S}_i\}_{i=1}^5$ whose bases $\{\mathbf{U}_i\}_{i=1}^5$ are $100 \times 4$ random matrices consisting of orthogonal columns ($\tau = 4$ and $r = 20$). We then sample 20 data from each subspace uniformly and obtain a $100 \times 100$ data matrix. The sample size $d$ varies from 1 to 100, and we record the empirical probability of success over 200 times of experiments, where we define that an algorithm succeeds if $\|\widehat{\mathbf{L}} - \mathbf{L}\|_F \leq 10^{-6}$ and $\mathbf{rank}(\widehat{\mathbf{L}}) = r$. As shown in Figure 1, we see that the sample complexity can indeed be smaller in the case of mixture of subspaces.

Figure 1: Comparison of sample complexity between the case of single subspace and that of mixture of subspaces.

## 2 Proof of Robust Recovery under Deterministic Noise

**Lemma 1** (Lemma 2. [1]). *Let* $\mathbf{W} = \mathbf{span}\{\mathbf{w}_1, \mathbf{w}_2, ..., \mathbf{w}_{k-1}\}$, $\mathbf{U} = \mathbf{span}\{\mathbf{w}_1, \mathbf{w}_2, ..., \mathbf{w}_{k-1}, \mathbf{u}\}$, *and* $\widetilde{\mathbf{U}} = \mathbf{span}\{\mathbf{w}_1, \mathbf{w}_2, ..., \mathbf{w}_{k-1}, \widetilde{\mathbf{u}}\}$ *be subspaces spanned by vectors in* $\mathbb{R}^m$. *Then*

$$\theta\left(\mathbf{U}, \widetilde{\mathbf{U}}\right) \leq \frac{\pi}{2} \frac{\theta(\widetilde{\mathbf{u}}, \mathbf{u})}{\theta(\widetilde{\mathbf{u}}, \mathbf{W})}.$$

**Lemma 2** (Lemma 2 in Main Body). *Let* $\mathbf{U}^k = \mathbf{span}\{\mathbf{u}_1, \mathbf{u}_2, ..., \mathbf{u}_k\}$ *and* $\widetilde{\mathbf{U}}^k = \mathbf{span}\{\widetilde{\mathbf{u}}_1, \widetilde{\mathbf{u}}_2, ..., \widetilde{\mathbf{u}}_k\}$ *be two subspaces such that* $\theta(\mathbf{u}_i, \widetilde{\mathbf{u}}_i) \leq \epsilon_{noise}$ *for all* $i \in [k]$. *Let* $\gamma_k = \sqrt{20k\epsilon_{noise}}$ *and* $\theta(\widetilde{\mathbf{u}}_i, \widetilde{\mathbf{U}}^{i-1}) \geq \gamma_i$ *for* $i = 2, ..., k$. *Then*

$$\theta\left(\mathbf{U}^k, \widetilde{\mathbf{U}}^k\right) \leq 10k\frac{\epsilon_{noise}}{\gamma_k} = \frac{\gamma_k}{2}. \tag{1}$$

*Proof.* The proof is basically by induction on $k$. Instead, we will prove a stronger result by showing that (1) holds on subspaces $\mathbf{U}^k = \mathbf{span}\{\mathbf{W}, \mathbf{u}_1, \mathbf{u}_2, ..., \mathbf{u}_k\}$ and $\widetilde{\mathbf{U}}^k = \mathbf{span}\{\mathbf{W}, \widetilde{\mathbf{u}}_1, \widetilde{\mathbf{u}}_2, ..., \widetilde{\mathbf{u}}_k\}$ for arbitrary fixed subspace $\mathbf{W}$. The base case $k = 1$ follows immediately from Lemma 1. Now suppose the conclusion holds for any index $\leq k - 1$. Let $\mathbf{U}_0^k = \mathbf{span}\{\mathbf{U}^{k-1}, \widetilde{\mathbf{u}}_k\}$. Then for the index $k$, we have

$$\begin{aligned}
\theta(\mathbf{U}^k, \widetilde{\mathbf{U}}^k) &\leq \theta(\mathbf{U}^k, \mathbf{U}_0^k) + \theta(\mathbf{U}_0^k, \widetilde{\mathbf{U}}^k) \\
&\leq \frac{\pi}{2} \frac{\theta(\widetilde{\mathbf{u}}_k, \mathbf{u}_k)}{\theta(\widetilde{\mathbf{u}}_k, \mathbf{U}^{k-1})} + 10(k-1)\frac{\epsilon_{noise}}{\gamma_{k-1}} \quad \text{(By Lemma 1 and induction hypothesis)} \\
&\leq \frac{\pi}{2} \frac{\epsilon_{noise}}{\theta(\widetilde{\mathbf{u}}_k, \widetilde{\mathbf{U}}^{k-1}) - \theta(\mathbf{U}^{k-1}, \widetilde{\mathbf{U}}^{k-1})} + 10(k-1)\frac{\epsilon_{noise}}{\gamma_{k-1}} \\
&\leq \frac{\pi}{2} \frac{\epsilon_{noise}}{\gamma_k - 10(k-1)\frac{\epsilon_{noise}}{\gamma_{k-1}}} + 10(k-1)\frac{\epsilon_{noise}}{\gamma_{k-1}} \quad \text{(By induction hypothesis)} \\
&= \frac{\epsilon_{noise}}{\gamma_{k-1}} \left( \frac{\pi}{2} \frac{\gamma_{k-1}^2}{\gamma_k \gamma_{k-1} - 10(k-1)\epsilon_{noise}} + 10(k-1) \right) \\
&\leq \frac{\epsilon_{noise}}{\gamma_{k-1}} (\pi + 10k - 10) \\
&= \frac{\epsilon_{noise}}{\gamma_k} \frac{\gamma_k}{\gamma_{k-1}} (\pi + 10k - 10) \\
&= \frac{\epsilon_{noise}}{\gamma_k} \sqrt{\frac{k}{k-1}} (\pi + 10k - 10) \\
&\leq \frac{\epsilon_{noise}}{\gamma_k} 10k \quad (k \geq 2).
\end{aligned}$$

$\square$

**Lemma 3** (Lemma 3 in Main Body). *Let* $\epsilon_k = 2\gamma_k$, $\gamma_k = \sqrt{20k\epsilon_{noise}}$, *and* $k \leq r$. *Suppose that we observe a set of coordinates* $\Omega \subset [m]$ *of size* $d$ *uniformly at random with replacement, where* $d \geq c_0(\mu_0 r + mk\epsilon_{noise}) \log^2(2/\delta)$. *If* $\theta(\mathbf{L}_{:t}, \widetilde{\mathbf{U}}^k) \leq \epsilon_k$, *then with probability at least* $1 - 4\delta$, *we have*

$$\|\mathbf{M}_{\Omega t} - \mathcal{P}_{\widetilde{\mathbf{U}}_{\Omega:}^k} \mathbf{M}_{\Omega t}\|_2 \leq C\sqrt{\frac{dk\epsilon_{noise}}{m}}.$$

*Inversely, if* $\theta(\mathbf{L}_{:t}, \widetilde{\mathbf{U}}^k) \geq c\epsilon_k$, *then with probability at least* $1 - 4\delta$, *we have*

$$\|\mathbf{M}_{\Omega t} - \mathcal{P}_{\widetilde{\mathbf{U}}_{\Omega:}^k} \mathbf{M}_{\Omega t}\|_2 \geq C\sqrt{\frac{dk\epsilon_{noise}}{m}},$$

*where* $c_0$, $c$ *and* $C$ *are absolute constants.*

*Proof.* The first part of the theorem follows from the upper bound of Lemma 15. Specifically, by plugging $d$ into the lower bound of Lemma 15, we see that $\alpha < 1/2$ and $\gamma < 1/3$. Note that

$$\left\|\mathbf{L}_{:t} - \mathcal{P}_{\widetilde{\mathbf{U}}^k}\mathbf{L}_{:t}\right\|_2 = \|\mathbf{L}_{:t}\|_2 \sin\theta\left(\mathbf{L}_{:t}, \mathcal{P}_{\widetilde{\mathbf{U}}^k}\mathbf{L}_{:t}\right) \leq \theta\left(\mathbf{L}_{:t}, \mathcal{P}_{\widetilde{\mathbf{U}}^k}\mathbf{L}_{:t}\right) = \theta(\mathbf{L}_{:t}, \widetilde{\mathbf{U}}^k) \leq \epsilon_k.$$

Therefore, by Lemma 15,

$$\left\| \mathbf{M}_{\Omega t} - \mathcal{P}_{\widetilde{\mathbf{U}}_{\Omega:}^k} \mathbf{M}_{\Omega t} \right\|_2 \leq \mathcal{O} \left( \sqrt{\frac{d}{m}} \left\| \mathbf{M}_{:t} - \mathcal{P}_{\widetilde{\mathbf{U}}^k} \mathbf{M}_{:t} \right\|_2 \right)$$

$$\leq \mathcal{O} \left( \sqrt{\frac{d}{m}} \left( \left\| \mathbf{L}_{:t} - \mathcal{P}_{\widetilde{\mathbf{U}}^k} \mathbf{L}_{:t} \right\|_2 + \left\| \mathcal{P}_{\widetilde{\mathbf{U}}^k} (\mathbf{L}_{:t} - \mathbf{M}_{:t}) \right\|_2 + \left\| \mathbf{M}_{:t} - \mathbf{L}_{:t} \right\|_2 \right) \right)$$

$$\leq \mathcal{O} \left( \sqrt{\frac{d}{m}} (\epsilon_k + 2\epsilon_{noise}) \right)$$

$$\leq C \sqrt{\frac{dk\epsilon_{noise}}{m}}.$$

We now proceed the second part of the theorem. To this end, we first explore the relation between the incoherence of the noisy basis $\widetilde{\mathbf{U}}^k$ and the clean one $\mathbf{U}^k$. Since we are able to control the error propagation in $\widetilde{\mathbf{U}}^k$, intuitively, the incoherence of $\widetilde{\mathbf{U}}^k$ and $\mathbf{U}^k$ is not distinct too much. In particular, for any $i \in [m]$,

$$\left\| \mathcal{P}_{\widetilde{\mathbf{U}}^k} \mathbf{e}_i \right\|_2 \leq \left\| \mathcal{P}_{\mathbf{U}^k} \mathbf{e}_i \right\|_2 + \left\| \mathcal{P}_{\mathbf{U}^k} \mathbf{e}_i - \mathcal{P}_{\widetilde{\mathbf{U}}^k} \mathbf{e}_i \right\|_2$$

$$\leq \left\| \mathcal{P}_{\mathbf{U}^k} \mathbf{e}_i \right\|_2 + \left\| \mathcal{P}_{\mathbf{U}^k} - \mathcal{P}_{\widetilde{\mathbf{U}}^k} \right\| \left\| \mathbf{e}_i \right\|_2$$

$$= \left\| \mathcal{P}_{\mathbf{U}^k} \mathbf{e}_i \right\|_2 + \left\| \mathbf{e}_i \right\|_2 \sin \theta \left( \mathbf{U}^k, \widetilde{\mathbf{U}}^k \right)$$

$$\leq \left\| \mathcal{P}_{\mathbf{U}^k} \mathbf{e}_i \right\|_2 + \theta \left( \mathbf{U}^k, \widetilde{\mathbf{U}}^k \right)$$

$$\leq \left\| \mathcal{P}_{\mathbf{U}^k} \mathbf{e}_i \right\|_2 + \frac{\gamma_k}{2}$$

$$= \left\| \mathcal{P}_{\mathbf{U}^k} \mathbf{e}_i \right\|_2 + \frac{1}{4}\epsilon_k.$$

Therefore,

$$\mu \left( \widetilde{\mathbf{U}}^k \right) = \frac{m}{k} \max_{i \in [m]} \left\| \mathcal{P}_{\widetilde{\mathbf{U}}^k} \mathbf{e}_i \right\|_2^2 \leq \frac{m}{k} \left( 2 \left\| \mathcal{P}_{\mathbf{U}^k} \mathbf{e}_i \right\|_2^2 + \frac{1}{8}\epsilon_k^2 \right) \leq 2\mu(\mathbf{U}^k) + c'' m \epsilon_{noise},$$

for global constant $c''$. Also, note that

$$\left\| \mathcal{P}_{\widetilde{\mathbf{U}}^k} \mathbf{M}_{:t} - \mathbf{L}_{:t} \right\|_2 \geq \sin \theta \left( \mathbf{L}_{:t}, \mathcal{P}_{\widetilde{\mathbf{U}}^k} \mathbf{M}_{:t} \right) \left\| \mathbf{L}_{:t} \right\|_2 \geq \frac{1}{2}\theta \left( \mathbf{L}_{:t}, \mathcal{P}_{\widetilde{\mathbf{U}}^k} \mathbf{M}_{:t} \right) \geq \frac{1}{2}\theta \left( \mathbf{L}_{:t}, \widetilde{\mathbf{U}}^k \right) \geq \frac{c\epsilon_k}{2}.$$

So we have

$$\left\| \mathbf{M}_{\Omega t} - \mathcal{P}_{\widetilde{\mathbf{U}}_{\Omega}^k} \mathbf{M}_{\Omega t} \right\|_2 \geq \sqrt{\frac{1}{m} \left( \frac{d}{2} - \frac{3k\mu(\widetilde{\mathbf{U}}^k)\beta}{2} \right)} \left\| \mathbf{M}_{:t} - \mathcal{P}_{\widetilde{\mathbf{U}}^k} \mathbf{M}_{:t} \right\|_2$$

$$\geq \Omega \left( \sqrt{\frac{d}{m} - \frac{3k\mu(\mathbf{U}^k)}{m} \log^2(1/\delta) - c_0 k\epsilon_{noise} \log^2(1/\delta)} \left\| \mathbf{M}_{:t} - \mathcal{P}_{\widetilde{\mathbf{U}}^k} \mathbf{M}_{:t} \right\|_2 \right)$$

$$\geq \Omega \left( \sqrt{\frac{d}{m} - \frac{3\mu_0 r}{m} \log^2(1/\delta) - c_0 k\epsilon_{noise} \log^2(1/\delta)} \left\| \mathbf{M}_{:t} - \mathcal{P}_{\widetilde{\mathbf{U}}^k} \mathbf{M}_{:t} \right\|_2 \right) \quad (\text{Since } \mathbf{U}^k \subseteq \mathbf{U}^r)$$

$$\geq \Omega \left( \sqrt{\frac{d}{m} - c_0 k\epsilon_{noise} \log^2(1/\delta)} \left( \left\| \mathcal{P}_{\widetilde{\mathbf{U}}^k} \mathbf{M}_{:t} - \mathbf{L}_{:t} \right\|_2 - \left\| \mathbf{L}_{:t} - \mathbf{M}_{:t} \right\|_2 \right) \right) \quad (\text{Since } d > 3\mu_0 r \log^2(1/\delta))$$

$$> \Omega \left( \sqrt{\frac{d}{m} - c_0 k\epsilon_{noise} \log^2(1/\delta)} \left( \frac{c\epsilon_k}{2} - \epsilon_{noise} \right) \right)$$

$$> C \sqrt{\frac{dk\epsilon_{noise}}{m}} \quad (\text{Since } d > c_0 m k\epsilon_{noise} \log^2(1/\delta)).$$

$\square$

---

**Algorithm 1** Noise-Tolerant Life-Long Matrix Completion under Bounded Deterministic Noise

---

**Input:** Columns of matrices arriving over time.

**Initialize:** Let the basis matrix $\widehat{\mathbf{U}}^0 = \emptyset$. Randomly draw entries $\Omega \subset [m]$ of size $d$ uniformly with replacement.

**1: For** $t$ from 1 to $n$, **do**

**2:**    (a) If $\|\mathbf{M}_{\Omega t} - \mathcal{P}_{\widehat{\mathbf{U}}^k_{\Omega:}} \mathbf{M}_{\Omega t}\|_2 > \eta_k$

**3:**       i. Fully measure $\mathbf{M}_{:t}$ and add it to the basis matrix $\widehat{\mathbf{U}}^k$. Orthogonalize $\widehat{\mathbf{U}}^k$.

**4:**       ii. Randomly draw entries $\Omega \subset [m]$ of size $d$ uniformly with replacement.

**5:**       iii. $k := k + 1$.

**6:**    (b) Otherwise $\widehat{\mathbf{M}}_{:t} := \widehat{\mathbf{U}}^k \widehat{\mathbf{U}}^{k\dagger}_{\Omega:} \mathbf{M}_{\Omega t}$.

**7: End For**

**Output:** Estimated range space $\widehat{\mathbf{U}}^K$ and the underlying matrix $\widehat{\mathbf{M}}$ with column $\widehat{\mathbf{M}}_{:t}$.

---

**Theorem 4** (Theorem 1 in Main Body). *Let $r$ be the rank of the underlying matrix $\mathbf{L}$ with $\mu_0$-incoherent column space. Suppose that the $\ell_2$ norm of the noise in each column is upper bounded by $\epsilon_{noise}$. Set $d \geq c_0(\mu_0 r + mk\epsilon_{noise})\log^2(2n/\delta))$ and $\eta_k = C\sqrt{dk\epsilon_{noise}/m}$ for some global constants $c_0$ and $C$. Then Algorithm 1 outputs $\widehat{\mathbf{U}}^K$ with $K \leq r$, $\widehat{\mathbf{M}}$ with $\ell_2$ error $\|\widehat{\mathbf{M}}_{:t} - \mathbf{L}_{:t}\|_2 \leq \Theta\left(\frac{m}{d}\sqrt{k\epsilon_{noise}}\right)$ uniformly for all $t$ with probability at least $1 - \delta$, where $k \leq r$ is the number of base vectors when learning the $t$-th column.*

*Proof.* We first show $K \leq r$. Every time we add a new direction to the basis matrix if and only if Condition (a) in Algorithm 1 holds true. In that case by Lemma 3, if setting $\eta_k = C\sqrt{dk\epsilon_{noise}/m}$, then with probability at least $1 - 4\delta$, we have that $\theta(\mathbf{L}_{:t}, \widetilde{\mathbf{U}}^k) \geq 2\gamma_k$, which implies $\theta(\mathbf{M}_{:t}, \widetilde{\mathbf{U}}^k) \geq \theta(\mathbf{L}_{:t}, \widetilde{\mathbf{U}}^k) - \theta(\mathbf{M}_{:t}, \mathbf{L}_{:t}) \geq \gamma_k$. So by Lemma 2, $\theta(\mathbf{U}^k, \widetilde{\mathbf{U}}^k) \leq \gamma_k/2$. Thus $\theta(\mathbf{L}_{:t}, \mathbf{U}^k) \geq \theta(\mathbf{L}_{:t}, \widetilde{\mathbf{U}}^k) - \theta(\mathbf{U}^k, \widetilde{\mathbf{U}}^k) \geq 3\gamma_k/2$. Since $\mathbf{rank}(\mathbf{L}) = r$, we obtain that $K \leq r$.

We now proceed to prove the upper bound on the $\ell_2$ error. We discuss Case (a) and (b) respectively. If Condition (a) in Algorithm 1 holds true, then according to the algorithm, we fully observe $\mathbf{M}_{:t}$ and use it as our estimate $\widehat{\mathbf{M}}_{:t}$. So $\left\|\widehat{\mathbf{M}}_{:t} - \mathbf{L}_{:t}\right\|_2 \leq \epsilon_{noise} \leq \Theta(\frac{m}{d}\sqrt{k\epsilon_{noise}})$; On the other hand, if Case (b) in Algorithm 1 holds true, then we represent $\widehat{\mathbf{M}}_{:t}$ by the basis subspace $\widetilde{\mathbf{U}}^k$. So we have

$$\left\|\widehat{\mathbf{M}}_{:t} - \mathbf{L}_{:t}\right\|_2 = \left\|\widetilde{\mathbf{U}}^k\widetilde{\mathbf{U}}^{k\dagger}_{\Omega:}\mathbf{M}_{\Omega t} - \mathbf{L}_{:t}\right\|_2$$

$$\leq \left\|\widetilde{\mathbf{U}}^k\widetilde{\mathbf{U}}^{k\dagger}\mathbf{L}_{:t} - \mathbf{L}_{:t}\right\|_2 + \left\|\widetilde{\mathbf{U}}^k\widetilde{\mathbf{U}}^{k\dagger}_{\Omega:}\mathbf{L}_{\Omega t} - \widetilde{\mathbf{U}}^k\widetilde{\mathbf{U}}^{k\dagger}\mathbf{L}_{:t}\right\|_2 + \left\|\widetilde{\mathbf{U}}^k\widetilde{\mathbf{U}}^{k\dagger}_{\Omega:}\mathbf{L}_{\Omega t} - \widetilde{\mathbf{U}}^k\widetilde{\mathbf{U}}^{k\dagger}_{\Omega:}\mathbf{M}_{\Omega t}\right\|_2$$

$$= \sin\theta(\mathbf{L}_{:t}, \widetilde{\mathbf{U}}^k) + \left\|\widetilde{\mathbf{U}}^k\widetilde{\mathbf{U}}^{k\dagger}_{\Omega:}\mathbf{L}_{\Omega t} - \widetilde{\mathbf{U}}^k\widetilde{\mathbf{U}}^{k\dagger}\mathbf{L}_{:t}\right\|_2 + \left\|\widetilde{\mathbf{U}}^k\widetilde{\mathbf{U}}^{k\dagger}_{\Omega:}(\mathbf{L}_{\Omega t} - \mathbf{M}_{\Omega t})\right\|_2.$$

To bound the second term, let $\mathbf{L}_{:t} = \widetilde{\mathbf{U}}^k\mathbf{v} + \mathbf{e}$, where $\widetilde{\mathbf{U}}^k\mathbf{v} = \widetilde{\mathbf{U}}^k\widetilde{\mathbf{U}}^{k\dagger}\mathbf{L}_{:t}$ and $\|\mathbf{e}\|_2 \leq \epsilon_k$ since $\|\mathbf{e}\|_2 = \sin\theta(\mathbf{L}_{:t}, \widetilde{\mathbf{U}}^k) \leq \epsilon_k$. So

$$\widetilde{\mathbf{U}}^k\widetilde{\mathbf{U}}^{k\dagger}_{\Omega:}\mathbf{L}_{\Omega t} - \widetilde{\mathbf{U}}^k\widetilde{\mathbf{U}}^{k\dagger}\mathbf{L}_{:t} = \widetilde{\mathbf{U}}^k(\widetilde{\mathbf{U}}^{kT}_{\Omega:}\widetilde{\mathbf{U}}^k_{\Omega:})^{-1}\widetilde{\mathbf{U}}^{kT}_{\Omega:}(\widetilde{\mathbf{U}}^k_{\Omega:}\mathbf{v} + \mathbf{e}_\Omega) - \widetilde{\mathbf{U}}^k_{\Omega:}\mathbf{v}$$

$$= \widetilde{\mathbf{U}}^k\widetilde{\mathbf{U}}^{k\dagger}_{\Omega:}\mathbf{e}_\Omega.$$

Therefore,

$$\left\|\widehat{\mathbf{M}}_{:t} - \mathbf{L}_{:t}\right\|_2 \leq \theta(\mathbf{L}_{:t}, \widetilde{\mathbf{U}}^k) + \left\|\widetilde{\mathbf{U}}^k\widetilde{\mathbf{U}}^{k\dagger}_{\Omega:}\mathbf{L}_{\Omega t} - \widetilde{\mathbf{U}}^k\widetilde{\mathbf{U}}^{k\dagger}\mathbf{L}_{:t}\right\|_2 + \left\|\widetilde{\mathbf{U}}^k\widetilde{\mathbf{U}}^{k\dagger}_{\Omega:}(\mathbf{L}_{\Omega t} - \mathbf{M}_{\Omega t})\right\|_2$$

$$\leq \theta(\mathbf{L}_{:t}, \widetilde{\mathbf{U}}^k) + \left\|\widetilde{\mathbf{U}}^k\widetilde{\mathbf{U}}^{k\dagger}_{\Omega:}\right\|\|\mathbf{e}_\Omega\|_2 + \left\|\widetilde{\mathbf{U}}^k\widetilde{\mathbf{U}}^{k\dagger}_{\Omega:}\right\|\|\mathbf{L}_{\Omega t} - \mathbf{M}_{\Omega t}\|_2$$

$$\leq \epsilon_k + \Theta\left(\frac{m}{d}\epsilon_k\right) + \Theta\left(\frac{m}{d}\epsilon_{noise}\right)$$

$$= \Theta\left(\frac{m}{d}\sqrt{k\epsilon_{noise}}\right),$$

where $\left\|\widetilde{\mathbf{U}}^k\widetilde{\mathbf{U}}^k_{\Omega:}\right\| \leq \sigma_1(\widetilde{\mathbf{U}}^k)/\sigma_k(\widetilde{\mathbf{U}}^k_{\Omega:}) \leq \Theta(m/d)$ once $d \geq \Omega(\mu(\widetilde{\mathbf{U}}^k)k\log(k/\delta))$, due to Lemma 16. The final sample complexity follows from the union bound on the $n$ columns. $\square$

# 3 Proof of Exact Recovery under Random Noise

---

**Algorithm 2** Noise-Tolerant Life-Long Matrix Completion under Sparse Random Noise

---

**Input:** Columns of matrices arriving over time.

**Initialize:** Let the basis matrix $\widehat{\mathbf{B}}^0 = \emptyset$, the counter $\mathbf{C} = \emptyset$. Randomly draw entries $\Omega \subset [m]$ of size $d$ uniformly without replacement.

1: **For** each column $t$ of $\mathbf{M}$, **do**
2:    (a) If $\|\mathbf{M}_{\Omega t} - \mathcal{P}_{\widehat{\mathbf{B}}^k_{\Omega:}} \mathbf{M}_{\Omega t}\|_2 > 0$
3:       i. Fully measure $\mathbf{M}_{:t}$ and add it to the basis matrix $\widehat{\mathbf{B}}^k$.
4:       ii. $\mathbf{C} := [\mathbf{C}, 0]$.
5:       iii. Randomly draw entries $\Omega \subset [m]$ of size $d$ uniformly without replacement.
6:       iv. $k := k + 1$.
7:    (b) Otherwise
8:       i. $\mathbf{C} := \mathbf{C} + \mathbf{1}^T_{supp(\widehat{\mathbf{B}}^{k\dagger}_{\Omega:} \mathbf{M}_{\Omega t})}$.    //Record supports of representation coefficient
9:       ii. $\widehat{\mathbf{M}}_{:t} := \widehat{\mathbf{B}}^k \widehat{\mathbf{B}}^{k\dagger}_{\Omega:} \mathbf{M}_{\Omega t}$.
10:    $t := t + 1$.
11: **End For**

**Outlier Removal:** Remove columns corresponding to entry 0 in vector $\mathbf{C}$ from $\widehat{\mathbf{B}}^{s_0+r} = [\mathbf{E}^{s_0}, \mathbf{U}^r]$.

**Output:** Estimated range space, identified outlier vectors, and recovered underlying matrix $\widehat{\mathbf{M}}$ with column $\widehat{\mathbf{M}}_{:t}$.

---

**Lemma 5.** *Let $\mathbf{X} = \mathbf{U}\mathbf{\Sigma}\mathbf{V}^T$ be the skinny SVD of $\mathbf{X}$, $\mathbf{orth}_c(\mathbf{X}) = \mathbf{U}$, and $\mathbf{orth}_r(\mathbf{X}) = \mathbf{V}^T$. Then for any set of coordinates $\Omega$ and any matrix $\mathbf{X} \in \mathbb{R}^{m \times n}$, we have*

$$\mathbf{rank}(\mathbf{X}_{\Omega:}) = \mathbf{rank}([\mathbf{orth}_c(\mathbf{X})]_{\Omega:}) \quad and \quad \mathbf{rank}(\mathbf{X}_{:\Omega}) = \mathbf{rank}([\mathbf{orth}_r(\mathbf{X})]_{:\Omega}).$$

*Proof.* Let $\mathbf{X} = \mathbf{U}\mathbf{\Sigma}\mathbf{V}^T$ be the skinny SVD of matrix $\mathbf{X}$, where $\mathbf{U} = \mathbf{orth}_c(\mathbf{X})$ and $\mathbf{V}^T = \mathbf{orth}_r(\mathbf{X})$. On one hand,

$$\mathbf{X}_{\Omega:} = \mathbf{I}_{\Omega:}\mathbf{X} = \mathbf{I}_{\Omega:}\mathbf{U}\mathbf{\Sigma}\mathbf{V}^T = [\mathbf{orth}_c(\mathbf{X})]_{\Omega:}\mathbf{\Sigma}\mathbf{V}^T.$$

So $\mathbf{rank}(\mathbf{X}_{\Omega:}) \le \mathbf{rank}([\mathbf{orth}_c(\mathbf{X})]_{\Omega:})$. On the other hand, we have

$$\mathbf{X}_{\Omega:}\mathbf{V}\mathbf{\Sigma}^{-1} = [\mathbf{orth}_c(\mathbf{X})]_{\Omega:}.$$

Thus $\mathbf{rank}([\mathbf{orth}_c(\mathbf{X})]_{\Omega:}) \le \mathbf{rank}(\mathbf{X}_{\Omega:})$. So $\mathbf{rank}(\mathbf{X}_{\Omega:}) = \mathbf{rank}([\mathbf{orth}_c(\mathbf{X})]_{\Omega:})$.

The second part of the argument can be proved similarly. Indeed, $\mathbf{X}_{:\Omega} = \mathbf{U}\mathbf{\Sigma}\mathbf{V}^T\mathbf{I}_{:\Omega} = \mathbf{U}\mathbf{\Sigma}[\mathbf{orth}_r(\mathbf{X})]_{:\Omega}$ and $\mathbf{\Sigma}^{-1}\mathbf{U}^T\mathbf{X}_{:\Omega} = [\mathbf{orth}_r(\mathbf{X})]_{:\Omega}$. So $\mathbf{rank}(\mathbf{X}_{:\Omega}) = \mathbf{rank}([\mathbf{orth}_r(\mathbf{X})]_{:\Omega})$, as desired. □

**Proposition 6.** *Let $\mathbf{L} \in \mathbb{R}^{m \times n}$ be any rank-$r$ matrix with skinny SVD $\mathbf{U}\mathbf{\Sigma}\mathbf{V}^T$. Denote by $\mathbf{L}_{:\Omega}$ the submatrix formed by subsampling the columns of $\mathbf{L}$ with i.i.d. Ber(d/n). If $d \ge 8\mu(\mathbf{V})r\log(r/\delta)$, then with probability at least $1 - \delta$, we have $\mathbf{rank}(\mathbf{L}_{:\Omega}) = r$. Similarly, denote by $\mathbf{L}_{\Omega:}$ the submatrix formed by subsampling the rows of $\mathbf{L}$ with i.i.d. Ber(d/m). If $d \ge 8\mu(\mathbf{U})r\log(r/\delta)$, then with probability at least $1 - \delta$, we have $\mathbf{rank}(\mathbf{L}_{\Omega:}) = r$.*

*Proof.* We only prove the first part of the argument. For the second part, applying the first part to matrix $\mathbf{L}^T$ gets the result. Denote by $\mathbf{T}$ the matrix $\mathbf{V}^T = \mathbf{orth}_r(\mathbf{L})$ with orthonormal rows, and by $\mathbf{X} = \sum_{i=1}^n \delta_i \mathbf{T}_{:i} e_i^T \in \mathbb{R}^{r \times n}$ the sampling of columns from $\mathbf{T}$ with $\delta_i \sim \text{Ber}(d/n)$. Let $\mathbf{X}_i = \delta_i \mathbf{T}_{:i} e_i^T$. Define positive semi-definite matrix

$$\mathbf{Y} = \mathbf{X}\mathbf{X}^T = \sum_{i=1}^n \delta_i \mathbf{T}_{:i}\mathbf{T}_{:i}^T.$$

Obviously, $\sigma_r^2(\mathbf{X}) = \lambda_r(\mathbf{Y})$. To invoke the matrix Chernoff bound, we estimate the parameters $L$ and $\mu_r$ in Lemma 16. Specifically, note that

$$\mathbb{E}\mathbf{Y} = \sum_{i=1}^n \mathbb{E}\delta_i \mathbf{T}_{:i}\mathbf{T}_{:i}^T = \frac{d}{n}\sum_{i=1}^n \mathbf{T}_{:i}\mathbf{T}_{:i}^T = \frac{d}{n}\mathbf{T}\mathbf{T}^T.$$

Therefore, $\mu_r = \lambda_r(\mathbb{E}\mathbf{Y}) = d\sigma_r^2(\mathbf{T})/n > 0$. Furthermore, we also have

$$\lambda_{\max}(\mathbf{X}_i) = \|\delta_i \mathbf{T}_{:i}\|_2^2 \leq \|\mathbf{T}\|_{2,\infty}^2 \triangleq L.$$

By the matrix Chernoff bound where we set $\epsilon = 1/2$,

$$\Pr[\sigma_r(\mathbf{X}) > 0] = \Pr[\lambda_r(\mathbf{Y}) > 0]$$

$$\geq \Pr\left[\lambda_r(\mathbf{Y}) > \frac{1}{2}\mu_r\right]$$

$$= \Pr\left[\lambda_r(\mathbf{Y}) > \frac{d}{2n}\sigma_r^2(\mathbf{T})\right]$$

$$\geq 1 - r\exp\left(-\frac{d\sigma_r^2(\mathbf{T})}{8n\|\mathbf{T}\|_{2,\infty}^2}\right)$$

$$\triangleq 1 - \delta.$$

So if

$$d \geq \frac{8n\|\mathbf{T}\|_{2,\infty}^2}{\sigma_r^2(\mathbf{T})}\log\frac{r}{\delta} = 8n\|\mathbf{T}\|_{2,\infty}^2\log\left(\frac{r}{\delta}\right),$$

then $\Pr[\sigma_{k+1}(\mathbf{X}) = 0] \leq \delta$, where the last equality holds since $\sigma_r(\mathbf{T}) = \sigma_r(\mathbf{V}^T) = 1$. Note that

$$\|\mathbf{T}\|_{2,\infty}^2 \leq \max_{i\in[n]}\|\mathbf{V}^T\mathbf{e}_i\|_2^2 \leq \frac{r}{n}\mu(\mathbf{V}).$$

So if $d \geq 8\mu(\mathbf{V})r\log(r/\delta)$ then with probability at least $1 - \delta$, $\mathbf{rank}(\mathbf{T}_{:\Omega}) = r$. Also, by Lemma 5, $\mathbf{rank}(\mathbf{T}_{:\Omega}) = \mathbf{rank}([\mathbf{orth}_r(\mathbf{L})]_{:\Omega}) = \mathbf{rank}(\mathbf{L}_{:\Omega})$. Therefore, $\mathbf{rank}(\mathbf{L}_{:\Omega}) = r$ with a high probability, as desired. $\qquad\square$

**Lemma 7.** *Let $\mathbf{U}^k \in \mathbb{R}^{m\times k}$ be a k-dimensional subspace of $\mathbf{U}^r$. Suppose we get access to a set of coordinates $\Omega \subset [m]$ of size d uniformly at random without replacement. Let $s \leq d - r - 1$ and $d \geq c\mu_0 r\log(k/\delta)$ for a universal constant c.*

1. *If $\mathbf{M}_{:t} \in \mathbf{U}^r$ but $\mathbf{M}_{:t} \notin \mathbf{U}^k$ then with probability at least $1-\delta$, $\mathbf{rank}\left([\mathbf{E}_{\Omega:}^s, \mathbf{U}_{\Omega:}^k, \mathbf{M}_{\Omega t}]\right) = s + k + 1$.*

2. *If $\mathbf{M}_{:t} \in \mathbf{U}^k$, then $\mathbf{rank}\left([\mathbf{E}_{\Omega:}^s, \mathbf{U}_{\Omega:}^k, \mathbf{M}_{\Omega t}]\right) = s + k$ with probability 1, the representation coefficients of $\mathbf{M}_{:t}$ corresponding to $\mathbf{E}^s$ in the dictionary $[\mathbf{E}^s, \mathbf{U}^k]$ is $\mathbf{0}$ with probability 1, and $[\mathbf{E}^s, \mathbf{U}^k][\mathbf{E}_{\Omega:}^s, \mathbf{U}_{\Omega:}^k]^\dagger\mathbf{M}_{\Omega t} = \mathbf{M}_{:t}$ with probability at least $1-\delta$.*

3. *If $\mathbf{M}_{:t} \notin \mathbf{U}^r$, i.e., $\mathbf{M}_{:t}$ is an outlier drawn from a non-degenerate distribution, then $\mathbf{rank}\left([\mathbf{E}_{\Omega:}^s, \mathbf{U}_{\Omega:}^k, \mathbf{M}_{\Omega t}]\right) = s + k + 1$ with probability $1 - \delta$.*

*Proof.* For the first part of the lemma, note that $\mathbf{rank}([\mathbf{U}^k, \mathbf{M}_{:t}]) = k + 1$. So according to Proposition 6, with probability $1-\delta$ we have that $\mathbf{rank}([\mathbf{U}^k, \mathbf{M}_{:t}]_{\Omega:}) = k+1$ since $d \geq c\mu_0 r\log((k+1)/\delta) \geq 8\mu([\mathbf{U}^k, \mathbf{M}_{:t}])k\log((k+1)/\delta)$ (Because $\mathbf{M}_{:t} \in \mathbf{U}^r$). Recall Facts 3 and 4 of Lemma 13 which imply that $\mathbf{rank}([\mathbf{E}^s, \mathbf{U}^k, \mathbf{M}_{:t}]_{\Omega:}) = s + k + 1$ when $s \leq d - r - 1$. This is what we desire.

For the middle part, the statement $\mathbf{rank}\left([\mathbf{E}_{\Omega:}^s, \mathbf{U}_{\Omega:}^k, \mathbf{M}_{\Omega t}]\right) = s + k$ comes from the assumption that $\mathbf{M}_{:t} \in \mathbf{U}^k$, which implies that $\mathbf{M}_{\Omega t} \in \mathbf{U}_{\Omega:}^k$ with probability 1, and that $\mathbf{rank}\left([\mathbf{E}_{\Omega:}^s, \mathbf{U}_{\Omega:}^k]\right) = s + k$ when $s \leq d - r - 1$ (Facts 3 and 4 of Lemma 13). Now suppose that the representation coefficients of $\mathbf{M}_{:t}$ corresponding to $\mathbf{E}^s$ in the dictionary $[\mathbf{E}^s, \mathbf{U}^k]$ is NOT $\mathbf{0}$ and $\mathbf{M}_{:t} \in \mathbf{U}^k$. Then $\mathbf{M}_{:t} - \mathbf{U}^k\mathbf{c} \in \mathbf{span}(\mathbf{E}^s)$, where $\mathbf{c}$ is the representation coefficients of $\mathbf{M}_{:t}$ corresponding to $\mathbf{U}^k$ in the dictionary $[\mathbf{E}^s, \mathbf{U}^k]$. Also, note that $\mathbf{M}_{:t} - \mathbf{U}^k\mathbf{c} \in \mathbf{U}^k$. So $\mathbf{rank}[\mathbf{E}^s, \mathbf{M}_{:t} - \mathbf{U}^k\mathbf{c}] = s$, which is contradictory with Fact 2 of Lemma 13. So the coefficient w.r.t. $\mathbf{E}^s$ in the dictionary $[\mathbf{E}^s, \mathbf{U}^k]$ is $\mathbf{0}$, and we have that $[\mathbf{E}^s, \mathbf{U}^k][\mathbf{E}_{\Omega:}^s, \mathbf{U}_{\Omega:}^k]^\dagger\mathbf{M}_{\Omega t} = \mathbf{U}^k\mathbf{U}_{\Omega:}^{k\dagger}\mathbf{M}_{\Omega t} = \mathbf{U}^k(\mathbf{U}_{\Omega:}^{kT}\mathbf{U}_{\Omega:}^k)^{-1}\mathbf{U}_{\Omega:}^{kT}\mathbf{M}_{\Omega t} = \mathbf{U}^k(\mathbf{U}_{\Omega:}^{kT}\mathbf{U}_{\Omega:}^k)^{-1}\mathbf{U}_{\Omega:}^{kT}\mathbf{U}_{\Omega:}^k\mathbf{v} = \mathbf{U}^k\mathbf{v} = \mathbf{M}_{:t}$, where $\mathbf{v}$ is the representation coefficient of $\mathbf{M}_{:t}$ w.r.t. $\mathbf{U}^k$. (The $(\mathbf{U}_{\Omega:}^{kT}\mathbf{U}_{\Omega:}^k)^{-1}$ exists because $\mathbf{rank}(\mathbf{U}_{\Omega:}^k) = k$ by Proposition 6)

As for the last part of the lemma, note that by Facts 2 and 4 of Lemma 13, $\mathbf{rank}([\mathbf{E}^s, \mathbf{M}_{:t}]_{\Omega:}) = s + 1$. Then by Fact 3 of Lemma 13 and the fact that $\mathbf{U}_{\Omega:}^k$ has rank $k$ (Proposition 6), we have $\mathbf{rank}\left([\mathbf{E}_{\Omega:}^s, \mathbf{U}_{\Omega:}^k, \mathbf{M}_{\Omega t}]\right) = s + k + 1$ when $s \leq d - r - 1$, as desired. $\qquad\square$

Now we are ready to prove Theorem 8.

**Theorem 8** (Theorem 4 in Main Body). *Let $r$ be the rank of the underlying matrix $\mathbf{L}$ with $\mu_0$-incoherent column space. Suppose that the outliers $\mathbf{E}^{s_0} \in \mathbb{R}^{m \times s_0}$ are drawn from any non-degenerate distribution, and that the underlying subspace $\mathbf{U}^r$ is identifiable. Then Algorithm 2 exactly recovers the underlying matrix $\mathbf{L}$, the column space $\mathbf{U}^r$, and the outlier $\mathbf{E}^{s_0}$ with probability at least $1 - \delta$, provided that $d \geq c\mu_0 r \log\left(\frac{r}{\delta}\right)$ and $s_0 \leq d - r - 1$. The total sample complexity is thus $c\mu_0 rn \log\left(\frac{r}{\delta}\right)$, where $c$ is a universal constant.*

*Proof.* The proof of Theorem 8 is an immediate result of Lemma 7 by using the union bound on the samplings of $\Omega$. Although Lemma 7 states that, for a specific column $\mathbf{M}_{:t}$, the algorithm succeeds with probability at least $1 - \delta$, the probability of success that uniformly holds for all columns is $1 - (r + s_0)\delta$ rather than $1 - n\delta$. This observation is from the proof of Lemma 7: $[\mathbf{E}^s, \mathbf{U}^k][\mathbf{E}_{\Omega:}^s, \mathbf{U}_{\Omega:}^k]^\dagger \mathbf{M}_{\Omega t} = \mathbf{M}_{:t}$ holds so long as $(\mathbf{U}_{\Omega:}^{kT}\mathbf{U}_{\Omega:}^k)^{-1}$ exists. Since in Algorithm 2 we resample $\Omega$ if and only if we add new vectors into the basis matrix, which happens at most $r + s_0$ times, the conclusion follows from the union bound of the $r + s_0$ events. Thus, to achieve a global probability of $1 - \delta$, the sample complexity for each upcoming column is $\Theta(\mu_0 r \log(r + s_0/\delta))$. Since we also require that $s_0 \leq d - r - 1$, the algorithm succeeds with probability $1 - \delta$ once $d \geq \Theta(\mu_0 r \log(d/\delta))$. Solving for $d$, we obtain that $d \gtrsim \mu_0 r \log(\mu_0^2 r^2/\delta^2) \asymp \mu_0 r \log(r/\delta)^1$. The total sample complexity for Algorithm 2 is thus $\Theta(\mu_0 rn \log(r/\delta))$.

For the exact identifiability of the outliers, we have the following guarantee:

**Lemma 9** (Outlier Removal). *Let the underlying subspace $\mathbf{U}^r$ be identifiable, i.e., for each $i \in [r]$, there are at least two columns $\mathbf{M}_{:a_i}$ and $\mathbf{M}_{:b_i}$ of $\mathbf{M}$ such that $[\mathbf{orth}_c(\mathbf{U}^r)]_{:i}^T\mathbf{M}_{:a_i} \neq 0$ and $[\mathbf{orth}_c(\mathbf{U}^r)]_{:i}^T\mathbf{M}_{:b_i} \neq 0$. Then the entries of $\mathbf{C}$ in Algorithm 2 corresponding to $\mathbf{U}^r$ cannot be 0's.*

*Proof.* Without loss of generality, let $\mathbf{U}^r$ be orthonormal. Suppose that the lemma does not hold true. Then there must exist one column $\mathbf{U}_{:i}^r$ of $\mathbf{U}^r$, say e.g., $\mathbf{e}_i$, such that $\mathbf{e}_i^T\mathbf{M}_{:t} = 0$ for all $t$ except when the index $t$ corresponds exactly to the $\mathbf{U}_{:i}^r$. This is contradictory with the condition that the subspace $\mathbf{U}^r$ is identifiable. The proof is completed. $\square$

Thus the proof of Theorem 8 is completed. $\square$

## 4 Proof of Lower Bound for Exact Recovery

**Theorem 10** (Theorem 5 in Main Body). *Let $0 < \delta < 1/2$, and $\Omega \sim \mathrm{Uniform}(d)$ be the index of the row sampling $\subseteq [m]$. Suppose that $\mathbf{U}^r$ is $\mu_0$-incoherent. If the total sampling number $dn < c\mu_0 rn \log(r/\delta)$ for a constant $c$, then with probability at least $1 - \delta$, there is an example of $\mathbf{M}$ such that under the sampling model of Section 2.1 in the main body (i.e., when a column arrives the choices are either (a) randomly sample or (b) view the entire column), there exist infinitely many matrices $\mathbf{L}'$ of rank $r$ obeying $\mu_0$-incoherent condition on column space such that $\mathbf{L}'_{\Omega:} = \mathbf{L}_{\Omega:}$.*

*Proof.* We prove the theorem by assuming that the underlying column space is known. Since we require additional samples to estimate the subspace, the proof under this assumption gives a lower bound. Let $\ell = \left\lfloor \frac{m}{\mu_0 r} \right\rfloor$. Construct the underlying matrix $\mathbf{L}$ by

$$\mathbf{L} = \sum_{k=1}^r b_k \mathbf{u}_k \mathbf{u}_k^T,$$

where the known $\mathbf{u}_k$ (Because the column space is known) is defined as

$$\mathbf{u}_k = \sqrt{\frac{1}{\ell}} \sum_{i \in B_k} \mathbf{e}_i, \quad B_k = \{(k-1)\ell + 1, (k-1)\ell + 2, ..., k\ell\}.$$

So the matrix $\mathbf{L}$ is a block diagonal matrix formulated as Figure 2. Further, construct the noisy matrix

Figure 2: Construction of underlying matrix $\mathbf{L}$.

$\mathbf{M}$ by $\mathbf{M} = [\mathbf{L}, \mathbf{E}]$. The matrix $\mathbf{E} \in \mathbb{R}^{m \times s_0}$ corresponds to the outliers, and the matrix $\mathbf{L}$ corresponds to the underlying matrix.

Notice that the information of $b_k$'s is only implied in the corresponding block of $\mathbf{L}$. So overall, the lower bound is given by solving from the inequality

$$\Pr\{\text{For all blocks, there must be at least one row being sampled}\} \geq 1 - \delta.$$

We highlight that the $b_k$'s can be chosen arbitrarily in that they do not change the coherence of the column space of $\mathbf{L}$. Also, it is easy to check that the column space of $\mathbf{L}$ is $\mu_0$-incoherent. By construction, the underlying matrix $\mathbf{L}$ is block-diagonal with $r$ blocks, each of which is of size $\ell \times \ell$. According to our sampling scheme, we always sample the same positions of the arriving column after the column space is known to us. This corresponds to sample the row of the matrix in hindsight. To recover $\mathbf{L}$, we argue that each block should have at least one row fully observed; Otherwise, there is no information to recover $b_k$'s. Let $A$ be the event that for a fixed block, none of its rows is observed. The probability $\pi_0$ of this event $A$ is therefore $\pi_0 = (1-p)^{\ell}$, where $p$ is the Bernoulli sampling parameter. Thus by independence, the probability of the event that there is at least one row being sampled holds true *for all diagonal blocks* is $(1 - \pi_0)^r$, which is $\geq 1 - \delta$ as we have argued. So

$$-r\pi_0 \geq r \log(1 - \pi_0) \geq \log(1 - \delta),$$

where the first inequality is due to the fact that $-x \geq \log(1 - x)$ for any $x < 1$. Since we have assumed $\delta < 1/2$, which implies that $\log(1 - \delta) \geq -2\delta$, thus $\pi_0 \leq 2\delta/r$. Note that $\pi_0 = (1-p)^{\ell}$, and so

$$-\log(1 - p) \geq \frac{1}{\ell} \log\left(\frac{r}{2\delta}\right) \geq \frac{\mu_0 r}{m} \log\left(\frac{r}{2\delta}\right).$$

This is equivalent to

$$mp \geq m\left(1 - \exp\left(-\frac{\mu_0 r}{m} \log \frac{r}{2\delta}\right)\right).$$

Note that $1 - e^{-x} \geq x - x^2/2$ whenever $x \geq 0$, we have

$$mp \geq (1 - \epsilon/2)\mu_0 r \log\left(\frac{r}{2\delta}\right),$$

where $\epsilon = \mu_0 r \log(r/2\delta) < 1$. Finally, by the equivalence between the uniform and Bernoulli sampling models (i.e., $d \approx mp$, Lemma 14), the proof is completed. $\qquad\square$

## 5 Mixture of Subspaces

In this section, we assume that the underlying subspace is a mixture of $h$ independent subspaces, each of which is of dimension at most $\tau \ll r$. Such an assumption naturally models settings in which there are really $h$ different categories of data while they share a certain commonality across categories. Indeed, many real datasets match the assumption, e.g., face image, 3D motion trajectory, handwritten digits, etc.

**Algorithm 3** Noise-Tolerant Life-Long Matrix Completion under Random Noise for Mixture of Subspaces

---

**Input:** Columns of matrices arriving over time.

**Initialize:** Let the basis matrix $\widehat{\mathbf{B}}^0 = \emptyset$, the counter $\mathbf{C} = \emptyset$. Randomly draw entries $\Omega \subset [m]$ of size $d$ uniformly without replacement.

**1: For** each column $t$ of $\mathbf{M}$, **do**

**2:**   (a) If there does not exist a $\tau$-sparse linear combination of columns of $\widehat{\mathbf{B}}^k_{\Omega:}$ that represents $\mathbf{M}_{\Omega t}$ exactly

**3:**     i. Fully measure $\mathbf{M}_{:t}$ and add it to the basis matrix $\widehat{\mathbf{B}}^k$.

**4:**     ii. $\mathbf{C} := [\mathbf{C}, 0]$.

**5:**     iii. Randomly draw entries $\Omega \subset [m]$ of size $d$ uniformly without replacement.

**6:**     iv. $k := k + 1$.

**7:**   (b) Otherwise

**8:**     i. $\mathbf{C} := \mathbf{C} + \mathbf{1}^T_{supp(\widehat{\mathbf{B}}^{k\dagger}_{\Omega:}\mathbf{M}_{\Omega t})}$.    //Record supports of representation coefficient

**9:**     ii. $\widehat{\mathbf{M}}_{:t} := \widehat{\mathbf{B}}^k \widehat{\mathbf{B}}^{k\dagger}_{\Omega:} \mathbf{M}_{\Omega t}$.

**10:**   $t := t + 1$.

**11: End For**

**Outlier Removal:** Remove columns corresponding to entry 0 in vector $\mathbf{C}$ from $\widehat{\mathbf{B}}^{s_0+r} = [\mathbf{E}^{s_0}, \mathbf{U}^r]$.

**Output:** Estimated range space, identified outlier vectors, and recovered underlying matrix $\widehat{\mathbf{M}}$ with column $\widehat{\mathbf{M}}_{:t}$.

---

**Lemma 11** (Mixture of Subspaces). *Let $[\mathbf{E}^s, \mathbf{U}^k]$ be the current dictionary matrix consisting of a random noise matrix $\mathbf{E}^s \in \mathbb{R}^{m \times s}$ and a clean basis matrix $\mathbf{U}^k \in \mathbb{R}^{m \times k}$. Suppose we get access to a set of coordinates $\Omega \subset [m]$ of size $d$ uniformly at random without replacement. Let $s \le d - \tau - 1$ and $d \ge 8\mu_\tau \tau \log(\tau/\delta)$. Denote by $\mathbf{U}^\tau \in \mathbb{R}^{m \times \tau}$ a submatrix of $\mathbf{U}^k$ with $\tau$ columns.*

1. *If $\mathbf{M}_{:t} \in \mathbf{U}^r$ but it cannot be represented by a linear combination of $\tau$ vectors in the current dictionary, then with probability at least $1 - \delta$, $\mathbf{M}_{\Omega t}$ does not belong to any fixed $\tau$-combination of the truncated dictionary as well.*

2. *If $\mathbf{M}_{:t}$ can be represented by a linear combination of $\tau$ vectors in the current basis, then $\mathbf{M}_{\Omega t}$ can be represented as a linear combination of the same $\tau$ truncated vectors in the dictionary with probability 1, the representation coefficients of $\mathbf{M}_{:t}$ corresponding to $\mathbf{E}^s$ in the dictionary is $\mathbf{0}$ with probability 1, and $[\mathbf{E}^s, \mathbf{U}^k][\mathbf{E}^s_{\Omega:}, \mathbf{U}^k_{\Omega:}]^\dagger \mathbf{M}_{\Omega t} = \mathbf{M}_{:t}$ with probability at least $1 - \delta$.*

3. *If $\mathbf{M}_{:t}$ is an outlier drawn from a non-degenerate distribution, then $\mathbf{M}_{\Omega t}$ cannot be represented by the dictionary with probability 1.*

*Proof.* The proof is similar as that of Lemma 7. For completeness, we give a brief proof here. For the first part of the lemma, by Facts 2 and 4 of Lemma 13, the $\mathbf{E}^s_{\Omega:}$ cannot have a non-zero representation coefficient of $\mathbf{M}_{\Omega t}$ in any possible $\tau$-combination of the current dictionary when $s \le d - \tau - 1$, due to the randomness. Thus the problem of whether $\mathbf{M}_{\Omega t}$ can be $\tau$ represented by the current dictionary $[\mathbf{E}^s_{\Omega:}, \mathbf{U}^k_{\Omega:}]$ is totally determined by whether it can be $\tau$ represented by $\mathbf{U}^k_{\Omega:}$. Now suppose that $\mathbf{M}_{\Omega t}$ can be written as a linear $\tau$-combination of the current basis $\mathbf{U}^\tau_{\Omega:}$. Then according to Proposition 6, since $d \ge 8\mu_\tau \tau \log(\tau/\delta)$, we have that $\mathbf{rank}([\mathbf{U}^\tau, \mathbf{M}_{:t}]) = \tau$, which is contradictory with the assumption of Event 1.

The first argument in Event 2 is obvious. Now suppose that the representation coefficients of $\mathbf{M}_{:t}$ corresponding to $\mathbf{E}^s$ in the dictionary $[\mathbf{E}^s, \mathbf{U}^k]$ is NOT $\mathbf{0}$ and $\mathbf{M}_{:t} \in \mathbf{U}^k$. Then $\mathbf{M}_{:t} - \mathbf{U}^k \mathbf{c} \in \mathbf{span}(\mathbf{E}^s)$, where $\mathbf{c}$ is the representation coefficients of $\mathbf{M}_{:t}$ corresponding to $\mathbf{U}^k$ in the dictionary $[\mathbf{E}^s, \mathbf{U}^k]$. Also, note that $\mathbf{M}_{:t} - \mathbf{U}^k \mathbf{c} \in \mathbf{U}^k$. So $\mathbf{rank}[\mathbf{E}^s, \mathbf{M}_{:t} - \mathbf{U}^k \mathbf{c}] = s$, which is contradictory with Fact 2 of Lemma 13. So the coefficient w.r.t. $\mathbf{E}^s$ in the dictionary $[\mathbf{E}^s, \mathbf{U}^k]$ is $\mathbf{0}$. Since by assumption $\mathbf{M}_{:t}$ can be represented by $\tau$ combination of columns in $\mathbf{U}^k$, termed $\mathbf{U}^\tau$, we have that $[\mathbf{E}^s, \mathbf{U}^k][\mathbf{E}^s_{\Omega:}, \mathbf{U}^k_{\Omega:}]^\dagger \mathbf{M}_{\Omega t} = \mathbf{U}^\tau \mathbf{U}^{\tau\dagger}_{\Omega:} \mathbf{M}_{\Omega t} = \mathbf{U}^\tau (\mathbf{U}^{\tau T}_{\Omega:} \mathbf{U}^\tau_{\Omega:})^{-1} \mathbf{U}^{\tau T}_{\Omega:} \mathbf{M}_{\Omega t} =$

$\mathbf{U}^\tau (\mathbf{U}_{\Omega:}^{\tau T} \mathbf{U}_{\Omega:}^\tau)^{-1} \mathbf{U}_{\Omega:}^{\tau T} \mathbf{U}_{\Omega:}^\tau \mathbf{v} = \mathbf{U}^\tau \mathbf{v} = \mathbf{M}_{:t}$, where $\mathbf{v}$ is the representation coefficient of $\mathbf{M}_{:t}$ w.r.t. $\mathbf{U}^\tau$. (The $(\mathbf{U}_{\Omega:}^{\tau T} \mathbf{U}_{\Omega:}^\tau)^{-1}$ exists because $\mathbf{rank}(\mathbf{U}_{\Omega:}^\tau) = \tau$ by Proposition 6)

Event 3 is an immediate result of Lemma 13. $\qquad\square$

**Theorem 12** (Theorem 6 in main body). *Let $r$ be the rank of the underlying matrix $\mathbf{L}$. Suppose that the columns of $\mathbf{L}$ lie on a mixture of independent subspaces, each of which is of dimension at most $\tau$. Denote by $\mu_\tau$ the maximal incoherence over all $\tau$-combinations of $\mathbf{L}$. Let the noise model be that of Theorem 8. Then Algorithm 3 exactly recovers the underlying matrix $\mathbf{L}$, the column space $\mathbf{U}^r$, and the outlier $\mathbf{E}^{s_0}$ with probability at least $1 - \delta$, provided that $d \geq c\mu_\tau \tau^2 \log\left(\frac{r}{\delta}\right)$ for some global constant $c$ and $s_0 \leq d - \tau - 1$. The total sample complexity is thus $c\mu_\tau \tau^2 n \log\left(\frac{r}{\delta}\right)$.*

*Proof.* Theorem 12 is a result of union bound of Lemma 11. For the event of type 1, the union bound is over $\binom{r}{\tau} = \mathcal{O}(r^\tau)$ events. For the event of type 2, since we resample $\Omega$ at most $r + s_0$ times by algorithm, the union bound is over $r + s_0$ samplings. The event of type 3 is with probability 1. So overall, replacing $\delta$ with $\min\{\delta/r^\tau, \delta/(r + s_0)\}$ in Lemma 11, the sample complexity we need is at least $\mathcal{O}(\mu_\tau \tau \log(\max\{r^\tau, r + s_0\}/\delta))$. Note that $s_0 \leq d - \tau - 1$. So the sample complexity for each column is at least $\mathcal{O}(\mu_\tau \tau^2 \log(r/\delta))$ and the total one is $\mathcal{O}(\mu_\tau \tau^2 n \log(r/\delta))$, as desired. The success of outlier removal step is guaranteed by Lemma 9. $\qquad\square$

## 6 Facts on Subspace Spanned by Non-Degenerate Random Vectors

**Lemma 13.** *Let $\mathbf{E}^s \in \mathbb{R}^{m \times s}$ be matrix consisting of corrupted vectors drawn from any non-degenerate distribution. Let $\mathbf{U}^k \in \mathbb{R}^{m \times k}$ be any fixed matrix with rank $k$. Then with probability 1, we have*

1. $\mathbf{rank}(\mathbf{E}^s) = s$ *for any $s \leq m$;*

2. $\mathbf{rank}([\mathbf{E}^s, \mathbf{x}]) = s + 1$ *holds for $\mathbf{x} \in \mathbf{U}^k \subset \mathbb{R}^m$ uniformly and $s \leq m - k$, where $\mathbf{x}$ can even depend on $\mathbf{E}^s$;*

3. $\mathbf{rank}([\mathbf{E}^s, \mathbf{U}^k]) = s + k$, *provided that $s + k \leq m$;*

4. *The marginal of non-degenerate distribution is non-degenerate.*

*Proof.* For simplicity, we only show the proof of Fact 1. The other facts can be proved similarly. Let $\mathbf{E}^s = [\mathbf{E}^{s-1}, \mathbf{e}]$. Since $\mathbf{e}$ is drawn from a non-degenerate distribution, the conditional probability satisfies $\Pr[\mathbf{rank}(\mathbf{E}^{s-1}, \mathbf{e}) = s \mid \mathbf{E}^{s-1}] = 1$ by the definition of non-degenerate distribution. So $\Pr[\mathbf{rank}(\mathbf{E}^{s-1}, \mathbf{e}) = s] = \mathbb{E}_{\mathbf{E}^{s-1}} \Pr[\mathbf{rank}(\mathbf{E}^{s-1}, \mathbf{e}) = s \mid \mathbf{E}^{s-1}] = 1$. $\qquad\square$

## 7 Equivalence between Bernoulli and Uniform Models

**Lemma 14.** *Let $n$ be the number of Bernoulli trials and suppose that $\Omega \sim Ber(d/n)$. Then with probability at least $1 - \delta$, $|\Omega| = \Theta(d)$, provided that $d \geq 4\log(1/\delta)$.*

*Proof.* Take a perturbation $\epsilon$ such that $d/n = d_0/n + \epsilon$. By the scalar Chernoff bound which states that
$$\Pr(|\Omega| \leq d_0) \leq e^{-\epsilon^2 n^2/2d_0},$$
if taking $d_0 = d/2$, $\epsilon = d/2n$ and $d \geq 4\log(1/\delta)$, we have
$$\Pr(|\Omega| \leq d/2) \leq e^{-d/4} \leq \delta. \tag{2}$$

In the other direction, by the scalar Chernoff bound again which states that
$$\Pr(|\Omega| \geq d_0) \leq e^{-\epsilon^2 n^2/3d},$$
if taking $d_0 = 2d$, $\epsilon = -d/n$ and $d \geq 4\log(1/\delta)$, we obtain
$$\Pr(|\Omega| \geq 2d) \leq e^{-d/3} \leq \delta. \tag{3}$$

Finally, according to (2) and (3), we conclude that $d/2 < |\Omega| < 2d$ with probability at least $1 - \delta$. $\qquad\square$

# 8    A Collection of Concentration Results

**Lemma 15** (Theorem 6. [3])**.** *Denote by $\widetilde{\mathbf{U}}^k$ a k-dimensional subspace in $\mathbb{R}^m$. Let the sampling number $d \geq \max\{\frac{8}{3}k\mu(\widetilde{\mathbf{U}}^k)\log(\frac{2k}{\delta}), 4\mu(\mathcal{P}_{\widetilde{\mathbf{U}}^{k\perp}}y)\log(\frac{1}{\delta})\}$. Denote by $\Omega$ an index set of size $d$ sampled uniformly at random with replacement from $[m]$. Then with probability at least $1 - 4\delta$, for any $y \in \mathbb{R}^m$, we have*

$$\frac{d(1-\alpha) - k\mu(\widetilde{\mathbf{U}}^k)\frac{\beta}{1-\zeta}}{m} \left\| y - \mathcal{P}_{\widetilde{\mathbf{U}}^k}y \right\|_2^2 \leq \left\| y_\Omega - \mathcal{P}_{\widetilde{\mathbf{U}}^k_{\Omega:}}y_\Omega \right\|_2^2 \leq (1+\alpha)\frac{d}{m} \left\| y - \mathcal{P}_{\widetilde{\mathbf{U}}^k}y \right\|_2^2,$$

*where $\alpha = \sqrt{2\frac{\mu(\mathcal{P}_{\widetilde{\mathbf{U}}^{k\perp}}y)}{d}\log(1/\delta)} + \frac{2\mu(\mathcal{P}_{\widetilde{\mathbf{U}}^{k\perp}}y)}{3d}\log(1/\delta)$, $\beta = (1 + 2\log(1/\delta))^2$, and $\zeta = \sqrt{\frac{8k\mu(\widetilde{\mathbf{U}}^k)}{3d}\log(2r/\delta)}$.*

**Lemma 16** (Matrix Chernoff Bound [2])**.** *Consider a finite sequence $\{\mathbf{X}_k\} \in \mathbb{R}^{n \times n}$ of independent, random, Hermitian matrices. Assume that*

$$0 \leq \lambda_{\min}(\mathbf{X}_k) \leq \lambda_{\max}(\mathbf{X}_k) \leq L.$$

*Define $\mathbf{Y} = \sum_k \mathbf{X}_k$, and $\mu_r$ as the r-th largest eigenvalue of the expectation $\mathbb{E}\mathbf{Y}$, i.e., $\mu_r = \lambda_r(\mathbb{E}\mathbf{Y})$. Then*

$$\Pr\{\lambda_r(\mathbf{Y}) > (1-\epsilon)\mu_r\} \geq 1 - r\left[\frac{e^{-\epsilon}}{(1-\epsilon)^{1-\epsilon}}\right]^{\frac{\mu_r}{L}} \geq 1 - re^{-\frac{\mu_r\epsilon^2}{2L}} \ for \ \epsilon \in [0,1).$$

## Footnotes

[1] We assume here that $\mu_0 \leq \mathrm{poly}(r/\delta)$.