[Reviews · NeurIPS 2016]

Reviewer 1

Summary

The paper proposes solutions to the streaming matrix completion problem, under various low rank and noise models, where full-state measurements can be acquired adaptively. The authors present both an algorithm and theoretical guarantees for matrix recovery. The paper is clearly written and the notation well chosen.

Qualitative Assessment

The technical work is very good and well presented. The paper describes an intuitive solution to a long-standing and impactful problem.

Confidence in this Review

2-Confident (read it all; understood it all reasonably well)


Reviewer 2

Summary

This paper considers the problem of estimating the column space of a low-rank matrix in a setting where given some random noisy samples from each column one can request to see the whole column, providing an algorithm that allows to reconstruct the matrix in the presence of noise and of corrupted columns with small sample complexity.

Qualitative Assessment

This paper considers the problem of estimating the column space of a low-rank matrix in a setting where given some random noisy samples from each column one can request to see the whole column, providing an algorithm that allows to reconstruct the matrix in the presence of noise and of corrupted columns with small sample complexity. I think that this is of interest, but unfortunately the way the paper is written is quite unclear and makes it difficult for a reader to understand its actual contributions. First, the paper focuses on the problem of online matrix recovery, but confusingly uses the term "life-long matrix completion" (which I have never encountered and actually yields zero results when entered in Google). The goal of lifelong learning is to: "develop life-long learning systems that learn many different tasks over time, and reuse insights from tasks learned" to quote from reference [1], but the paper considers a single task (estimating a low-rank matrix with a fixed column space from online samples). Second, the paper assumes that it is possible to request whole columns of the matrix on demand. This is what is meant by "adaptive" in the title. The authors should justify that this assumption is relevant to some real application (they just say that "in many real applications like Netflix rating prediction, the company can sequentially send out questionnaires to each person for movies, each of which corresponds to actively measure a column in full"; it is obviously impossible for Netflix to require a person to watch all of its movies or to expect to find out the opinion of every user about a specific movie) or suggest how the model could be modified to be relevant to some real application. In addition, due to the differences in the sampling models I don't think that it makes a lot of sense to compare the sample complexity of the algorithm to that of algorithms that truly use random undersampling and claim "state of the art results".

Confidence in this Review

2-Confident (read it all; understood it all reasonably well)


Reviewer 3

Summary

This paper studies recovering an incomplete low rank matrix with noise, in an online fashion. The model is such that the learner are allowed to adaptively choose to unifomrly sample d features, or do a full measurement. Two noise models are studied, namely the bounded deterministic noise model, and the sparse random noise model. The authors provide recovery guarantees for both cases (robust recovery for the first case and exact recovery for the second case). The result matches or improved the state-of-art. However, notice that results in literature are for passive learning, where the authors assumes active learning/adaptive sampling, so the comparison is just illustrative.

Qualitative Assessment

This paper studies recovering an incomplete low rank matrix with noise, in an online fashion. The model is such that the learner are allowed to adaptively choose to unifomrly sample d features, or do a full measurement. Two noise models are studied, namely the bounded deterministic noise model, and the sparse random noise model. The authors provide recovery guarantees for both cases (robust recovery for the first case and exact recovery for the second case). The result matches or improved the state-of-art. However, notice that results in literature are for passive learning, where the authors assumes active learning/adaptive sampling, so the comparison is just illustrative. This paper is well written and organized nicely. The contribution is in generally interesting. My main conerns are (1) whether the adaptive sampling model used is significant/practical. The authors motivate this model via Netflix rating prediction where the company can sequentially send out questionnaries to each person for movies. While this is a valid rational if the adaptive sampling involves a moderately higher sampling rate, it does not justify the proposed algorithm where one may request a column being full observed (equivalent to ask a user of Netflix to watch all movies and rate!!) (2) The sparse random noise model, where the noise is assumed to be sparse and iid following a distribution. In literature, most sparse noise are modeled as arbitrary and adversarial, and hence the model studied by the authors are much easier. Even in such case, the tolerable noise is very low (indeed the cardinality of the support of the noise must be lower than the cardinality of the observed entries). It is not clear to me how much this can be improved. The lower bound presented does not cover this issue either. Some minor comments: 1. subsubsection 3.1.2 should be a subsection, and 3.1.3, 3.1.4 combined should be a separate subsection too. 2. To avoid being misleading, the authors may want to mention that the lower bound does not rule out the possibility of improving the tolerance of the noise. 3. In theorem 6, the result depends on h very mildly, is there any intuition behind it?

Confidence in this Review

2-Confident (read it all; understood it all reasonably well)


Reviewer 4

Summary

This paper considers the problem of estimating or completing an ostensibly large, low-rank matrix whose columns present sequentially over time. The authors describe and analyze algorithmic approaches in two noise models - bounded adversarial and sparse column-wise corruptions, and also present extensions to settings where the low rank matrix manifests as a "mixture" of columns from lower rank matrices whose constituent columns collectively are drawn from independent subspaces. Experimental results are presented to validate the main results.

Qualitative Assessment

The problem(s) are well-defined and well (enough) motivated, and the results when presented are augmented with an appropriate level of intuitive explanation of their essence(s). Comparisons with existing methods might be made more clear if the requisite conditions on (in)coherence and sample complexity were compared e.g., in a table. This could also serve as an "at a glance" reference for the salient features of these methods over the prior works. The mixture of subspaces extension is interesting, and the results, perhaps, somewhat surprising. It might also be interesting to include in the main paper an experimental evaluation of this result on synthetic data and/or phase transitions, to elucidate the sample complexity improvement established in the corresponding result (Thm 6). In Figure 4, is the "estimated error" the error predicted by the theoretical result? If so, would an appropriate choice of the leading constant help to align the curves?

Confidence in this Review

2-Confident (read it all; understood it all reasonably well)


Reviewer 5

Summary

This paper analyzes a previously proposed online matrix completion algorithm that allows adaptive sampling, i.e., allows the user to select how many samples to observe for every new appearing column. The authors provide guarantees for the algorithm in two noisy settings - the bounded deterministic noise setting and in the sparse random noise setting. According to the authors, their sample complexity bounds are comparable to the best existing ones even though their method is online. Of course this is possible because their method allows adaptive sampling - if the projection of the new observed vector into the existing subspace estimate is large, the algorithm is allowed to acquire the entire matrix column.

Qualitative Assessment

This paper analyzes a previously proposed online matrix completion algorithm that allows adaptive sampling, i.e., allows the user to select how many samples to observe for every new appearing column. The authors provide guarantees for the algorithm in two noisy settings - the bounded deterministic noise setting and in the sparse random noise setting. According to the authors, their sample complexity bounds are comparable to the best existing ones even though their method is online. Of course this is possible because their method allows adaptive sampling - if the projection of the new observed vector into the existing subspace estimate is large, the algorithm is allowed to acquire the entire matrix column. The paper seems interesting although this reviewer has been unable to verify all the details due to the following missing/incomplete information. - In Algorithm 1, do you sample the set \Omega again at each time t or do you use the same set of all times t - Theorem 1: what is k? This should be clarified since the lower bound on d seems to depend on k. - It seems k is an integer equal to 1 or more. Then Sample complexity d > m k \epsilon_noise is meaningful only when epsilon << 1/k. Else you get a lower bound of d > m where m is the column length and that is meaningless. This fact should be mentioned - The authors say that, for the noise-free case their sample complexity bounds are comparable to the best ones; but they do not compare with the noisy MC literature results even though the focus of this paper is noisy online MC - Also what is the error bound on the subspace estimate \hat{U}^K? - The part on sparse random noise is hard to understand. Even the algorithm is not clear to this reviewer. The authors can have a better paper if they exclude the mixture of subspaces result while spend more time explaining the sparse random noise algorithm and theorem.

Confidence in this Review

2-Confident (read it all; understood it all reasonably well)


Reviewer 6

Summary

The paper studies the problem of online low rank matrix completion. The authors stated that their designed algorithms achieve strong guarantee under two noise models, say bounded deterministic noise and sparse random noise. Several experiments are conducted to verify the theoretical results.

Qualitative Assessment

1. Compared with [5,6], this works discards \mu(V) for incoherence \mu_0, which can not be determined as samples arrive one by one. From the recovery perspective, this issue makes the proposed technique limited to some special noises, like the deterministic and sparse RANDOM noises mentioned in the paper, and thus hard to be applied to real-world tasks. 2. Figure 1 is uninformative and unnecessary. The authors could provide more numerical experiments to demonstrate the advance of the proposed algorithms, since the given experiments are a bit insufficient and unconvincing. 3. Below are several related works: [1] Online Matrix Completion and Online Robust PCA, B.Lois and N. Vaswani, http://arxiv.org/abs/1503.03525, 2015 [2] Recursive Robust PCA or Recursive Sparse Recovery in Large but Structured Noise, C.Qiu, N. Vaswani, B. Lois and L. Hogben, IEEE Trans. Information Theory, 2014 [3] Online Robust Low Rank Matrix Recovery, X. Guo, IJCAI, 2015 [4] Online Robust PCA via Stochastic Optimization, J. Feng, H. Xu, and S. Yan, NIPS, 2013

Confidence in this Review

1-Less confident (might not have understood significant parts)